# Control of compound leaf patterning by *MULTI-PINNATE LEAF1* (*MPL1*) in chickpea

Ye Liu [1,2], Yuanfan Yang[2,3], Ruoruo Wang[2,4], Mingli Liu[2,5], Xiaomin Ji[2,4], Yexin He[2], Baolin Zhao[2], Wenju Li[2,5], Xiaoyu Mo[2,4], Xiaojia Zhang[2], Zhijia Gu[6], Bo Pan[7], Yu Liu[2], Million Tadege[8] ✉, Jianghua Chen [1,2,4,5] ✉ & Liangliang He [2,4] ✉

Plant lateral organs are often elaborated through repetitive formation of developmental units, which progress robustly in predetermined patterns along their axes. Leaflets in compound leaves provide an example of such units that are generated sequentially along the longitudinal axis, in species-specific patterns. In this context, we explored the molecular mechanisms underlying an acropetal mode of leaflet initiation in chickpea pinnate compound leaf patterning. By analyzing naturally occurring mutants *multi-pinnate leaf1* (*mpl1*) that develop higher-ordered pinnate leaves with more than forty leaflets, we show that *MPL1* encoding a C2H2-zinc finger protein sculpts a morphogenetic gradient along the proximodistal axis of the early leaf primordium, thereby conferring the acropetal leaflet formation. This is achieved by defining the spatiotemporal expression pattern of *CaLEAFY*, a key regulator of leaflet initiation, and also perhaps by modulating the auxin signaling pathway. Our work provides novel molecular insights into the sequential progression of leaflet formation.

Compound leaves, that consist of multiple independent units called leaflets, show wide diversity in patterning, ranging from trifoliolate, palmate, pinnate to higher-ordered forms[1]. Such variations have raised questions about the molecular mechanisms underlying their pattern formation[2]. A compound leaf begins as a simple peg-like primordium at the flanks of the shoot apical meristem (SAM)[3]. Then, initiation of leaflet primordia is dependent on the maintenance of a transient morphogenetic activity around the lateral margins of the primary primordium, and it occurs successively in either an acropetal (from proximal to distal) or basipetal (from distal to proximal) direction along the longitudinal axis[4–7]. Many morphogenetic regulators have

been identified and shown to be expressed in precise proximodistal patterns. In *Cardamine hirsuta*, which depends on *KNOXI* genes to maintain the relevant morphogenetic activity for leaflet initiation and exhibits a basipetal pattern of leaflet initiation, *ChSTM* and *ChBP1* show a proximal-to-distal decreasing gradient expression pattern in young leaf rachis[8–10]. Compound leaf development in tomato also follows a basipetal pattern, such that the distal leaflets differentiate earlier than the proximal leaflets. A *CIN-TCP* gene *LANCEOLATE* (*LA*) displays a distal-to-proximal decreasing gradient of expression, consistent with its role in promoting the basipetal differentiation of leaflets, and this expression pattern was complementary to the pattern of its cognate

[1]Division of Life Sciences and Medicine, School of Life Sciences, University of Science and Technology of China, Hefei, Anhui 230027, China. [2]CAS Key Laboratory of Topical Plant Resources and Sustainable Use, State Key Laboratory of Plant Diversity and Specialty Crops, Xishuangbanna Tropical Botanical Garden, Chinese Academy of Sciences, Kunming, Yunnan 650223, China. [3]School of Ecology and Environmental Sciences, Yunnan University, Kunming, Yunnan 650500, China. [4]University of Chinese Academy of Sciences, Beijing, China. [5]College of Life Science, Southwest Forestry University, Kunming, China. [6]CAS Key Laboratory for Plant Diversity and Biogeography of East Asia, Kunming Institute of Botany, Chinese Academy of Sciences, Kunming, Yunnan 650201, China. [7]Center for Integrative Conservation, Xishuangbanna Tropical Botanical Garden, Chinese Academy of Sciences, Mengla, Yunnan 666303, China. [8]Department of Plant and Soil Sciences, Institute for Agricultural Biosciences, Oklahoma State University, Ardmore, OK 73401, USA. ✉e-mail: million.tadege@okstate.edu; jhchen@xtbg.ac.cn; heliangliang@xtbg.ac.cn

*miR319* which has been shown to maintain the morphogenetic characteristics of leaf primordia[11]. As leaflets and tendrils in pea (*Pisum sativum*) leaves follow an acropetal pattern of initiation, the *LEAFY* (*LFY*) ortholog *UNIFOLIATA* (*UNI*) gene, which maintains the morphogenetic activity for the initiation of leaflet and tendril primordia, displays a distal-to-proximal descending gradient in early leaf primordia[12,13]. In pea mutants having more leaflets, the *UNI* expression pattern along the proximodistal axis showed dramatical changes[13]. Thus, there is agreement that the proximodistal expression patterns of morphogenetic regulators are associated with the compound leaf patterning. However, the mechanisms by which such patterns are established, maintained, and regulated, as well as their functional relationships with the sequential progression of leaflet formation are largely unknown despite improved knowledge of the molecular basis for the compound leaf development in tomato and *C. hirsuta*[14–18].

Legumes (Fabaceae), the third largest family of seed plants, typically have compound leaves[19]. The family is currently divided into six subfamilies: Cercidoideae, Detarioideae, Duparquetioideae, Dialioideae, Caesalpinioideae and Papilionoideae, with most having pinnate compound leaves in a variety of forms (except for Cercidoideae) (Fig. 1a)[20]. For instance, unipinnate leaves can be either paripinnate (when ending in two leaflets) or imparipinnate (when ending in one leaflet) with alternate or opposite leaflet arrangements (Fig. 1a). Pinnate leaves can also be bipinnate or tripinnate depending on the existence of second or third order leaflets, or some of the leaflets can even be modified into tendrils (Fig. 1a). Leaf development in legumes has mainly been studied in the trifoliolate leaves of *Medicago truncatula* and mungbean (*Vigna radiata*)[21–26], as well as the pinnate leaves of *Lotus japonicus* and pea which have some unique morphological and developmental characteristics themselves[12,13,27,28]. *Lotus japonicus* leaves have basal stipule-like leaflets and initiates their leaflets basipetally, while pea leaves have distal tendrils (Fig. 1a). Chickpea (*Cicer arietinum*), a very important pulse crop in the world[29,30], on the other hand, produces typical imparipinnate leaves that share several characteristics with many legumes, including all leaflets having almost uniform morphology and a strictly acropetal pattern of leaflet initiation[19]. In this study, by characterization of the naturally occurring mutants *multi-pinnate leaf1* (*mpl1*) of chickpea that have been known in the literature for more than 60 years[31,32], we uncover a morphogenetic gradient along the proximodistal axis of the primary primordium that maintains the acropetal pattern of leaflet formation, imparting chickpea its peculiar pinnately compound leaf pattern. Our work provides novel molecular insights into the sequential progression of leaflet formation, which is an important but not well-studied aspect of compound leaf development.

## Results

### Compound leaf development in chickpea
Leaf development in chickpea is heteroblastic similar to other legumes (Supplementary Fig. 1a). After the cotyledons open, two small and simple leaves known as "juvenile leaf" develop on the first and second nodes; each successive leaf becomes increasingly compound until an adult imparipinnate structure achieved around the eighth node (Fig. 1b; Supplementary Fig. 1a). A typical compound leaf consists of a pair of asymmetrically sized stipules (St) located at the proximal end of the petiole, 5–7 pairs of lateral leaflets (LL) arranged alternately along the rachis, and a terminal leaflet (TL) at the distal end (Fig. 1b).

We investigated the early ontogeny of chickpea imparipinnate leaves using scanning electron microscopy (SEM). At the plastochron 1 (P1) stage, a dome-shaped protrusion emerges at the flanks of the SAM (Fig. 1c, d), known as the compound leaf primordium (CLP). When the CLP reaches a height of ~50 μm (P2), a pair of stipule primordia (St) form on each side of its proximal end, encircling the shoot apex like a collar (Fig. 1c). During the P3 stage, when the CLP elongates to ~100 μm long, 2–3 pairs of LL primordia emerge consecutively as rounded protuberances from the base towards the tip along its margin (Fig. 1c, d). During the P4 stage, the tip of the CLP elongates progressively, and additional pairs of LL primordia continue to form along the lateral sides in an acropetal sequence (Fig. 1d–f). By late P4, when the primordium reaches a height of about 400 μm, the tip ceases its own elongation and differentiates into a TL primordium as indicated by development of trichomes from the abaxial surface (Fig. 1g). At this point, all of the LL primordia have completed their initiation and no new LL primordia are formed thereafter. At later stages, leaflet primordia became folded to the adaxial surfaces, followed by serration formation and rachis expansion (Fig. 1h–l). Trichomes, as a maker of cell differentiation, firstly emerge in late P4 from the abaxial surface of the TL primordium and later gradually develop on the stipule, LL primordia, petiole and rachis (Fig. 1e,g–l). Up to P8 stage, trichomes fairly distribute on the leaflet adaxial surfaces, marking the completion of the major configuration of the pinnate compound leaf (Fig. 1m).

### Isolation and characterization of *multi-pinnate leaf1* mutants of chickpea
By screening the USDA collection, we isolated a naturally occurring mutant line *multi-pinnate leaf1-1* (*mpl1-1*)(PI587041)[33], which exhibits a multi-pinnate compound leaf phenotype. Leaf forms of *mpl1-1* on nodes 1–4 closely resembled that of WT, but from node 5 onwards gradually became more complex than WT (Fig. 2a–c; Supplementary Fig. 1b). By node 10, it stabilized into an adult multi-pinnate form with 2–3 orders of over 40 leaflets, three times more than the WT (Fig. 2b–d). In such a leaf form, 5–7 pairs of first-order LLs are alternately arranged along the main axis (rachis) which ends in a TL, while numerous higher-order LLs alternate along the proximal lateral axes of the leaf, with their number decreasing gradually from the proximal to distal part (Fig. 2d, e). However, the number of first-order LLs was nearly equal to the number of LLs in WT leaves (Fig. 2f). These results indicated that the *mpl1* mutation converts normal LLs to compound leaf-like structures, with an effect of decreasing gradient from proximal to distal.

Compared to the ovate leaflet in WT leaves, *mpl1-1* leaves have lanceolate-shaped leaflets with severely serrated margins (Fig. 2g). These leaflets showed largely arrested expansion in both the transverse and longitudinal directions (Fig. 2g, h). The size of mature leaf epidermal cells was indistinguishable between *mpl1-1* and WT (Fig. 2i–k), but *mpl1-1* leaves have a smaller overall area (Fig. 2l), indicating that reduced cell proliferation rather than cell expansion contributed to the smaller and narrower leaflets in *mpl1*.

SEM analysis showed no difference between WT and *mpl1-1* leaf primordia during the early developmental stages from P1 to P4 (Fig. 2m, n; Supplementary Fig. 2a, b). Noticeable differences were first observed at the early P5 stage. At this point, compared to the folded LL primordia in WT, the first-order LL primordia of *mpl1-1* were rod-like in shape and the most proximal pairs initiated several bulges on their basal-margins acropetally, which represent the incipient second-order LL primordia (Fig. 2o, p; Supplementary Fig. 2c). Later, second-order LL primordia continued to emerge acropetally along the proximodistal axes of the first-order LL primordia in *mpl1-1* (Fig. 2q, r; Supplementary Fig. 2d). Leaflet primordia in WT developed into expanding blades at the P6 stage, while all leaflet primordia in *mpl1-1* remained slender in shape until the P8 stage, by which time they expanded a little (Fig. 2s–t; Supplementary Fig. 2d–f). These results indicate that CLPs in *mpl1-1* showed a prolonged and enhanced morphogenetic activity in their proximal regions but a retarded expansion of the leaflet blade compared to that of WT.

### *MPL1* encodes a C2H2 zinc-finger transcription factor
To clone the mutation, we conducted a cross between the *mpl1-1* mutant (♀) and the WT (cv. ICCV96029) (♂). The F1 plants display a WT phenotype and the F2 population segregated in a 3:1 ratio of WT plants

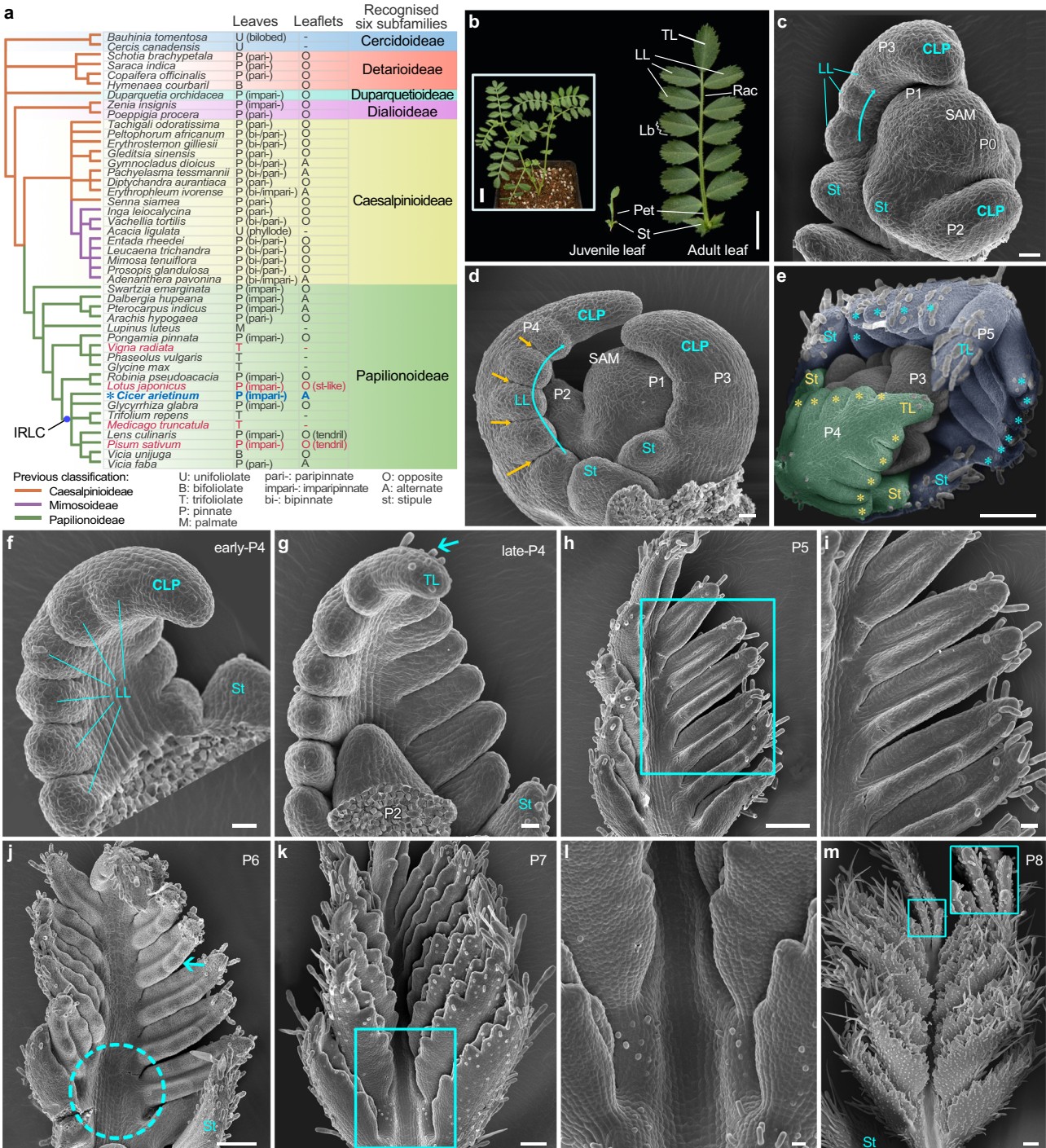

**Fig. 1 | The ontogeny of compound leaf development in wild type *Cicer arietinum* (chickpea). a** A new subfamily classification of the Leguminosae is derived from the NCBI taxonomy browser and Azani et al. (2017). The right table summarizes the leaf patterns and leaflet arrangements. The species highlighted in red indicate those that have received relatively extensive study regarding compound leaf development. **b** Morphology of a representative 6-week-old WT chickpea (cv. CDC Frontiers) (inset), a juvenile leaf (middle) and a mature leaf (right). Scale bars, 1 cm and 2 cm (inset). **c–m** Morphology of compound leaf primordia at different developmental stages. Images of shoot apical meristem (SAM) with three (**c**) or four (**d**) visible leaf primordia showing lateral leaflet (LL) initiation proceeded acropetally (cyan curved arrows) along lateral margins of compound leaf primordia (CLP) and boundaries established between LLs (yellow arrowheads). SAM with five visible

leaf primordia (P1-P5) (**e**) showing trichomes scare at the abaxial surface of the P4 leaf primordium (false-colored in green) but abundant at that of the P5 primordium (blue) with asterisks indicating LL primordia. Adaxial views of leaf primordia showing the distal portion of the CLP differentiated into a terminal leaflet (TL) (cyan arrowhead) primordium during the P4 stage (**f, g**), the leaflet primordia became folded to adaxial surfaces at the P5 stage (**h, i**), serration formation (cyan arrowhead) and rachis expansion (dotted cyan circle) at the P6 stage (**j**), the rachis elongation with trichomes forming on its adaxial surface at the P7 stage (**k, l**) and trichomes abundant on adaxial surfaces of leaflets (inset) at the P8 stage (**m**). **i, l** Close-up views of (**h** and **k**). Scale bars, 20 μm in (**c, d, f, g, i** and **l**), and 100 μm in (**e, h, j, k** and **m**). Similar results were obtained from three biological replicates for each tissue or organ.

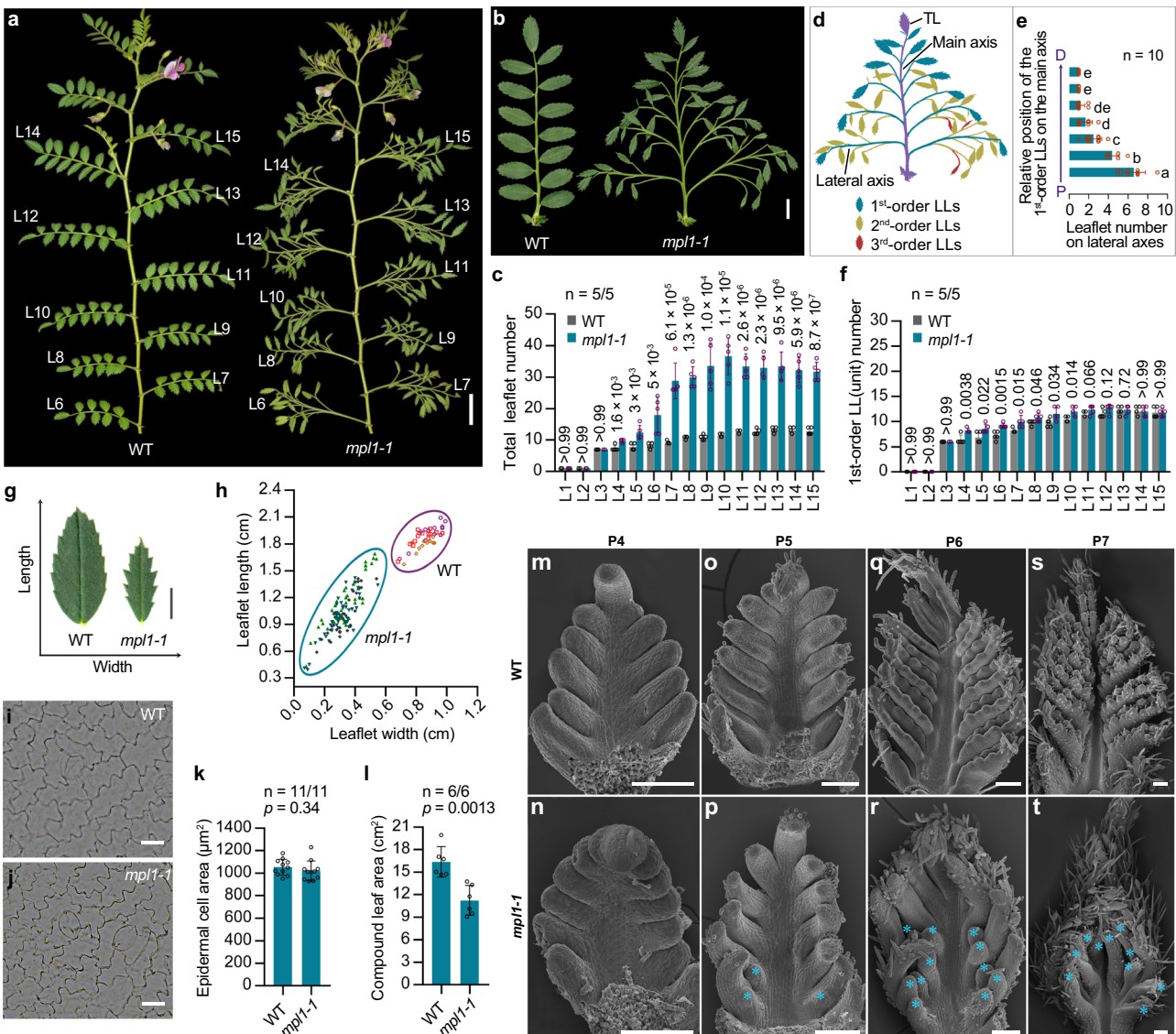

**Fig. 2 | Phenotypic comparison of WT and *multiple-pinnate leaf1-1* (*mpl1-1*) mutant of chickpea. a** Image of 10-week-old WT (cv. CDC Frontiers) and *mpl1-1* (PI587041) plants. L6–L15, leaf nodes 6–15, numbered from the cotyledon upward. **b** Mature compound leaves at the L12 node. **c** Quantification of total leaflet number in compound leaves from nodes L1–L15. Data shows mean ± SD from 5 plants. The values above bars represent the *p*-value estimated by the two-sided unpaired Student's *t* test for the comparison between WT and the *mpl1-1* mutant. **d** Illustration showing multiple orders of leaflet organization in a representative *mpl1-1* leaf. **e** Number of second- and third-order leaflets on lateral axes in a compound leaf. The x axis represents the leaflet number; the y axis represents the relative location of the lateral axes (the first-order LLs) along the main proximodistal axis (the main rachis). Data shows mean ± SD of 10 leaves. Different letters on the right of each bar indicate significant differences using the two-sided unpaired Student's *t* test (*p* < 0.05). **f** Quantification of the 1st-order lateral leaflet (LL) number in compound leaves from nodes L1–L15. Data shows mean ± SD from 5 plants. The values above bars represent the *p*-value estimated by the two-sided unpaired Student's *t* test for the comparison between WT and the *mpl1-1* mutant. **g** Representative leaflets dissected from WT and *mpl1-1* leaves. **h** Measurements of the leaflet length and width. Each label indicates one leaflet, and leaflets from a same compound are marked by same labels. Three mature compound leaves from the L12 node were measured for each genotype. Representative images of adaxial epidermal cells of mature WT (**i**) and *mpl1-1* (**j**) leaflets. **k** Average area of adaxial epidermal cells of WT and *mpl1-1* leaflets. Data shows mean ± SD of 11 cells. **l** Measurements of the blade area of compound leaves at the 12th node. Data shows mean ± SD of 6 leaves. The values above columns in (**k** and **l**) represent the *p*-value estimated using the two-sided unpaired Student's *t* test. SEM images of compound leaf primordia at the P4 (**m**, **n**), P5 (**o**, **p**), P6 (**q**, **r**) and P7 (**s**, **t**) stages. Cyan asterisk indicates the 2nd-order LL primordia in *mpl1-1*. Similar results were obtained from three biological replicates for each tissue or organ. Scale bars, 1 cm in (**a**), 2 cm in (**b**), 20 μm in (**i**, **j**), 100 μm in (**m–t**). Source data for Fig. 2c, e, f, h, k, l are provided as a Source Data file.

to *mpl1-1* mutants (163 WT and 52 *mpl1-1*) (Supplementary Fig. 3), indicating that *mpl1* is a single recessive mutation. Then, we performed a bulked segregant analysis (BSA), and found a peak associated with the *mpl1* phenotype located at a region on chromosome 8 (1.86–3.97 Mb, Fig. 3a, b; Supplementary Fig. 4). After filtered the SNPs and indels according specific criteria (see Methods) in the candidate region, we found five genes harboring genomic variants, out of which four genes had SNP variations, and one gene (*Ca_02268*) simultaneously had frameshift deletion and SNP variation (Fig. 3b; Supplementary Data 1). By analyzing genome annotation, performing BLAST analysis and considering the potential impact of deleterious variants on protein function, we identified *Ca_02268* as the most likely candidate gene. This locus was annotated as a protein consisting of a C2H2 Zinc-Finger domain at the N-terminus and a DLXLXLX-type EAR transcriptional repressor motif at the C-terminus (Fig. 3c), with high sequence similarity to the *M. truncatula* PALM1 (HM038482, NCBI). Loss-of-function *palm1* mutation is known to lead to the development of an extra pair of leaflets developed in the lateral leaflet regions[34]. This

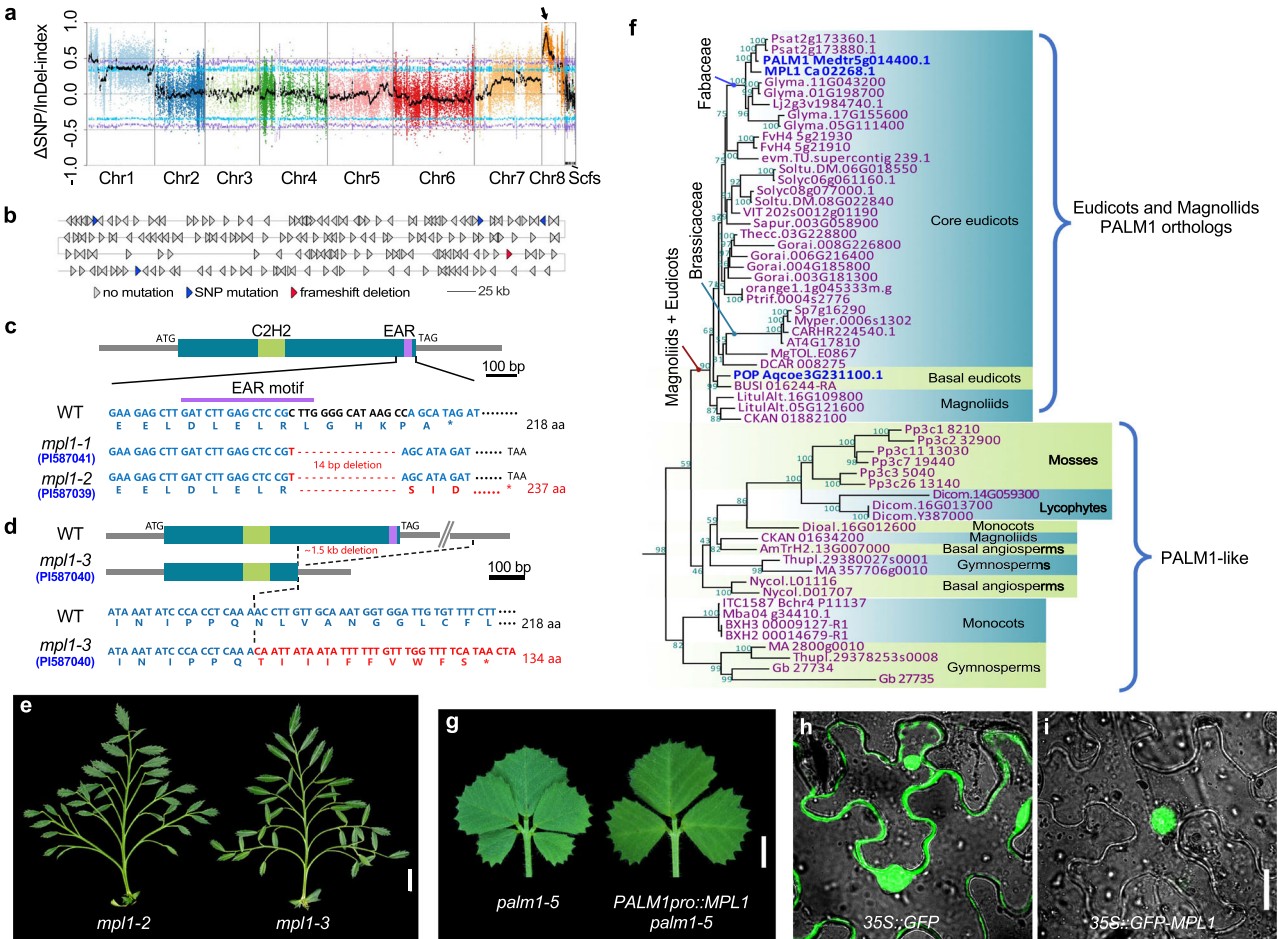

**Fig. 3 | Molecular cloning and characterization of *MPL1*. a** BSA analysis of F2 population derived from *mpl1-1* (PI587041) (♀)×WT (cv. ICCV96029) (♂). X-axis shows chromosomes/scaffolds of reference genome, Y-axis is ΔSNP/InDel index between WT and *mpl1* pools. Purple and blue lines indicate two-sided 99% and 95% confidence intervals. Black line is mean ΔSNP/InDel index with arrowhead indicating peak above threshold on Chromosome 8. **b** Enlargement of the peak of the BSA mapping interval. Each triangle represents an annotated gene, with blue indicating genes carrying SNP mutations and red indicating the most likely candidate gene *Ca_02268* that simultaneously carried SNP and deletion mutation. **c, d** Schematic diagram of genomic variations of *MPL1* in WT and *mpl1* alleles. *MPL1* gene, just one exon shown as dark-cyan box, with green and purple boxes representing the C₂H₂ Zinc-Finger domain and EAR motif, respectively. The *mpl1-1* and *mpl1-2* alleles had the same mutation (**c**), consisting of a base substitution (red) and

a 14 bp deletion (red dotted lines) within the *MPL1* gene, while the *mpl1-3* allele carried a 1.5 kb deletion encompassing a portion of the *MPL1* gene (**d**). **e** Mature compound leaves at the L12 node. **f** Phylogeny of MPL1 and its homologs from other species, constructed using the maximum-likelihood method and bootstrap test with 2000 replicates. Numbers on nodes represent bootstrap values. MPL1 and its functionally characterized orthologs of *Medicago truncatula* and *Aquilegia coerulea* are marked in blue. **g** Rescued the *M. truncatula palm1* mutant phenotype by *PALM1pro::MPL1*. Shown are representative compound leaves of *palm1-5* and an *PALM1pro::MPL1 palm1-5* transgenic line. Subcellular localization of *35S::GFP* (**h**) and *35S::GFP-MPL1* (**i**) transiently expressed in tobacco epidermal cells. Three independent experiments were performed with similar results. Scale bars, 1 cm in (**e**), 0.5 cm in (**g**), and 10 μm in (**h** and **i**).

is consistent in *Ca_02268* being the most likely candidate gene, and we designated it as *MPL1*.

PCR-based genotyping and sequencing analysis indicated that the *mpl1-1* mutant contains one base substitution (C to T) and 14 bp deletion at the 3' end of the *MPL1* gene, causing a frameshift mutation that affected the EAR transcriptional repressor motif (Fig. 3c; Supplementary Fig. 5a). Specifically, the mutation affects the EAR transcriptional repressor motif by removing a highly conserved leucine (L) residue in the DLELRSI sequence (Supplementary Fig. 5b), which is considered to be essential for maintaining the protein repressive activity[35]. Co-segregation analysis in an BC₂F₂ population indicated that 45 out of a total of 189 individuals displayed the *mpl1-1* mutant phenotype were homozygous for the deletion in *MPL1* (Supplementary Fig. 6a, b). Furthermore, another two *mpl1* mutant alleles isolated from the USDA, *mpl1-2* (PI587039) and *mpl1-3* (PI587040)[33], display a similar multi-pinnate leaf phenotype, also had mutations in *MPL1* (Fig. 3c–e). The *mpl1-2* had the same mutation as the *mpl1-1* (Fig. 3c), while the

*mpl1-3* carried a 1.5 kb deletion encompassing a portion of the *MPL1* gene that created a premature stop codon (Fig. 3d; Supplementary Fig. 7). Collectively, these results confirm that the loss-of-function mutation in *MPL1* is responsible for the *mpl1* mutant phenotype.

### *MPL1* is the orthologous gene of *PALM1*

Phylogenetic analysis of MPL1 homologs in land plants revealed that MPL1, PALM1, and POP from *Aquilegia coerulea* together form a distinct clade closely related to the SUPERMAN (SUP) and RABBIT EARS (RBE) clades (Supplementary Fig. 8a, b). The PALM1/POP/MPL1 proteins are highly conserved in eudicots and magnoliids, while show relatively greater divergence in other land plants, and are even lost in some lineages, such as rice and maize (Fig. 3f; Supplementary Fig. 8b). MPL1 and PALM1 both play critical roles in regulating compound leaf patterning, while POP is crucial for the development of floral nectar spurs in *Aquilegia* and also contributes to compound leaf development[36]. These data suggest that PALM1/POP/MPL1 have a

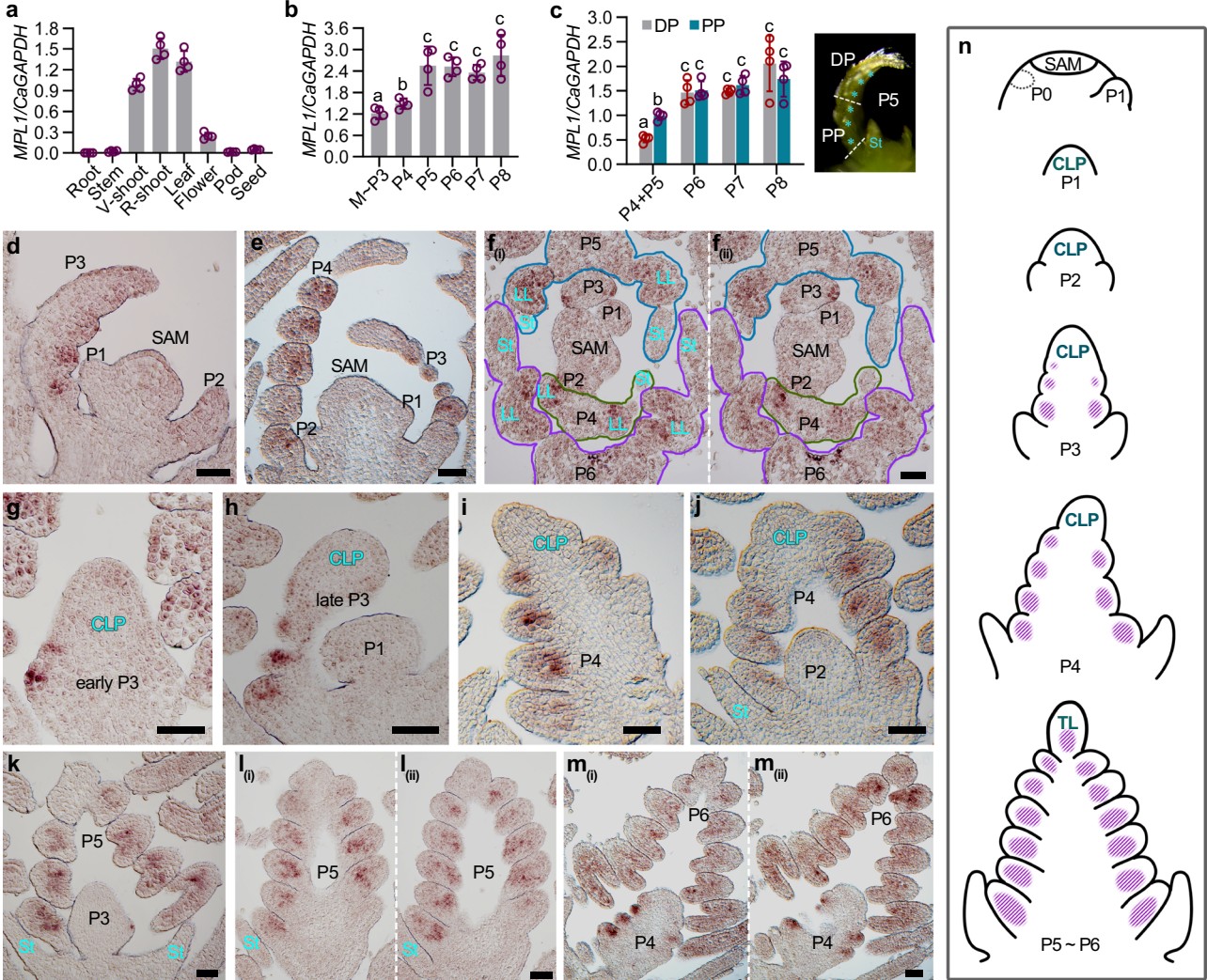

**Fig. 4 | The spatio-temporal expression pattern of *MPL1* during compound leaf development of chickpea. a** RT–qPCR analysis of *MPL1* mRNA expression levels in different tissues of WT (cv. CDC Frontiers) plants. **b** RT–qPCR analysis of *MPL1* mRNA expression levels at different developmental stages. M - P3: SAM with P1-P3 leaf primordia; P4-P8: leaf primordia at indicated developmental stages. **c** RT-qPCR analysis of *MPL1* mRNA expression levels in different portions of the P4-P8 leaf primordia. The different leaf portions are illustrated to the right with the dotted line showing the site of dissection. PP, Proximal Portion; DP, Distal Portion. Cyan asterisk indicates the LL primordia. Data shows mean ± SD of 4 biological replicates in (**a**–**c**). The value above columns in (**a**) represents the *p*-value estimated using the two-sided unpaired Student's *t* test. Different letters above bars in (**b** and **c**) indicate significant differences using the two-sided unpaired Student's *t* test ($p < 0.05$). RNA in situ hybridization of *MPL1*. Shown are single longitudinal (**d**, **e**) and two serial transverse (**f**) sections of the SAM with several leaf primordia, and single (**g**–**k**) and serial (**l**, **m**) longitudinal sections of the leaf primordia at the indicated developmental stages. Similar results were obtained from three independent experiments. St, Stipule; LL: Lateral leaflet; TL: Terminal leaflet. Scale bars, 50 μm. **n** Diagrams of compound leaf primordia at successive stages of ontogeny and the spatio-temporal expression pattern of *MPL1* (purple). Source data for Fig. 4a–c are provided as a Source Data file.

common and ancient origin in eudicots, with an ancestral function in leaf development.

The conserved role of *MPL1* in compound leaf development was confirmed by introducing a plasmid containing *MPL1* under the control of a 5-kb promoter of *PALM1* into *M. truncatula palm1* mutants (Supplementary Fig. 9a, b). Three independent transgenic lines showed a fully rescued leaf phenotype (Fig. 3g), and genotyping and RT-PCR analysis confirmed consistent expression of *MPL1* in these plants (Supplementary Fig. 9c-e). Subcellular localization assays revealed that MPL1 is localized to the nucleus, similar to PALM1 (Fig. 3h, i). These findings suggest that MPL1 and PALM1 are functional orthologs.

**Expression pattern of *MPL1* during compound leaf development**
To elucidate the precise role of *MPL1* in compound leaf development, we performed a series of experiments to clarify its expression pattern. RT-qPCR analysis in different tissues revealed that *MPL1* was highly

expressed in shoots and young leaves and moderately in flowers (Fig. 4a). A closer analysis of leaf development indicated that *MPL1* expression was moderate during the stages of leaflet initiation (from P3 to P4), significantly increased at the P5 stage, at which time point the whole primordium completely proceeded into differentiation, and maintained at a high level during later stages (from P6 to P8) (Fig. 4b). We also observed that *MPL1* expression was significantly higher in the proximal portion (PP) than the distal portion (DP) in the relatively young primordia (P4–P5) (Fig. 4c). However, in the more differentiated primordia (P6–P8), although the *MPL1* expression was increased compared to younger primordia, but there was no significant difference between the PP and DP (Fig. 4c).

We next studied the spatio-temporal pattern of *MPL1* expression during leaf development by RNA in situ hybridization. In a series of longitudinal and cross sections of the vegetative shoot apex, *MPL1* expression was clearly detected in leaf primordia at later stages

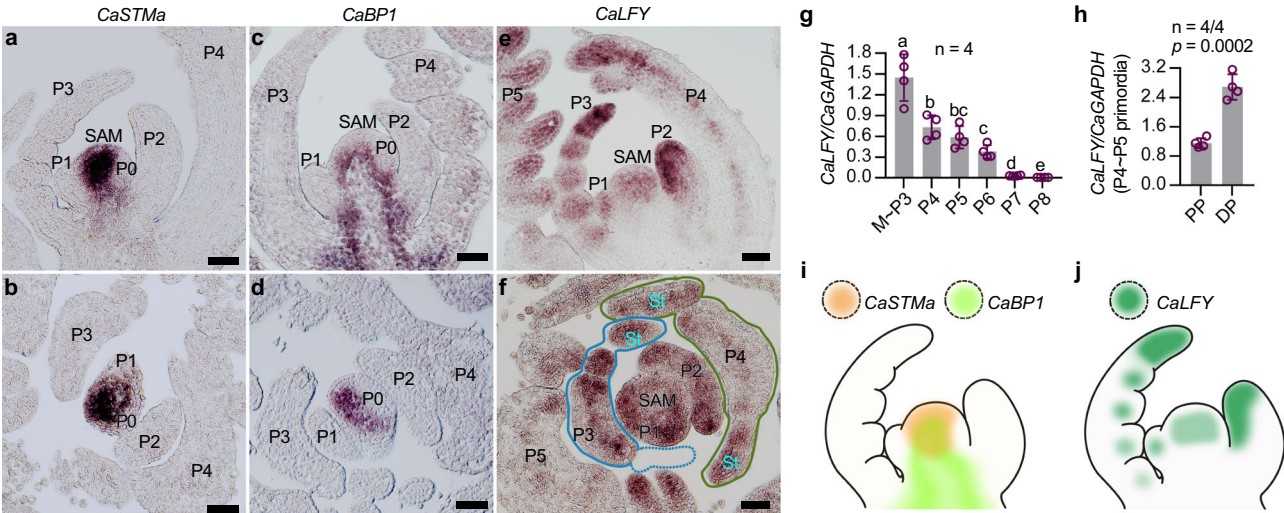

**Fig. 5 | Expression patterns of *KNOXI* genes and *CaLFY* in the vegetative shoot meristem of chickpea.** RNA in situ hybridization with *CaSTMa* (**a**, **b**), *CaBP1*(**c**, **d**) or *CaLFY* (**e**, **f**) specific probes on longitudinal (**a**, **c**, **e**) and transverse (**b**, **d**, **f**) sections of 4-week-old WT vegetative-shoots. Similar results were obtained from three independent experiments. Scale bars, 50 μm. **g** RT–qPCR analysis of *CaLFY* mRNA expression levels at different leaf developmental stages. Data shows mean ± SD of 4 biological replicates. Different letters above bars indicate significant differences using the two-sided unpaired Student's *t* test (*p* < 0.05). **h** RT–qPCR analysis of *CaLFY* expression in different portions of the P5 leaf primordia of WT. PP, Proximal Portion; DP, Distal Portion. Data shows mean ± SD of 4 biological replicates. The value above column represents the *p*-value estimated by the two-sided unpaired Student's *t* test. **i** A schematic illustration of *CaSTMa* and *CaBP1* expression pattern in the shoot apex. **j** A schematic illustration of *CaLFY* expression pattern in the shoot apex. Source data for Fig. 5g, h are provided as a Source Data file.

(P3–P6), predominantly in their LL primordia, but barely visible in the SAM, the P1–P2 leaf primordia and all stipule primordia (Fig. 4d–f; Supplementary Fig. 10a, b). *MPL1* expression was initially detected in the LL primordia at the P3 stage (Fig. 4d; Supplementary Fig. 10a). Upon initiation of LL primordia from the most base of the CLP, they immediately exhibited a robust *MPL1* expression (Fig. 4g). Subsequently, as LL primordia sequentially emerged from the lateral sides of the CLP, *MPL1* expression proceeds acropetally from the proximal LL primordia to the distal ones (Fig. 4h–j; Supplementary Fig. 10b–e). However, the undifferentiated tip of the whole CLP maintained an extremely weak or undetectable expression (Fig. 4h–j). During later stages, after all LL primordia completed their initiation and the tip of the CLP differentiated into the TL primordium, *MPL1* was broadly expressed in all LL primordia, and extended to the TL primordium (Fig. 4k–m; Supplementary Fig. 10f–h).

Overall, *MPL1* expression pattern is tightly associated with the sequential development of LL primordia. As the leaf develops, the *MPL1* expression in leaflet primordia proceeds progressively from the most-basal LL primordia to the distal TL primordium (Fig. 4n).

### Expression patterns of *KNOXI* genes and *CaLFY* in shoot apices of chickpea

The inverted repeat-lacking clade (IRLC) legumes, like pea and *M. truncatula* (Fig. 1a), depend on *LFY* genes to maintain the morphogenetic activity for leaflet initiation, while *KNOXI* genes are not associated[12,19,37,38]. Loss-of-function mutants of the *FLO/LFY* orthologs *SGL1* in *M. truncatula* and *UNI* in pea completely develop simple-like leaves, whereas those mutants with increased leaflet production showed great upregulation in *LFY* genes[12,13,23,37]. As chickpea is also an IRLC legume (Fig. 1a), we examined whether it relies on *CaLFY* but not *KNOXI* to regulate the morphogenetic activity. The *KNOXI* genes *CaSTMa* and *CaBP1* are expressed only in the SAM and excluded from all leaf primordia (Fig. 5a–d; Supplementary Fig. 11a–d), while *CaLFY* maintains a basal expression level in SAM but a strong expression in early leaf primordia (Fig. 5e, f). In these primordia, *CaLFY* was expressed along a proximal-distal gradient with a strongest signal in the undifferentiated tip of the leaf primordia, and it was also detected in the stipule primordia (Fig. 5e, f; Supplementary Fig. 12a, b). RT–qPCR

analysis revealed that the *CaLFY* was expressed at a highest level in the sample containing SAM and P1–P3 primordia, and then gradually reduced along with leaf differentiation and maturation (Fig. 5g). In line with the RNA in situ hybridization results, the *CaLFY* expression level in distal portion of the leaf primordia (P4–P5) was 2-folds higher than that of the proximal portion (Fig. 5h). These results indicate that the *CaLFY* rather than *KNOXI* was associated with the morphogenetic activity of the leaf primordia in chickpea, and the *CaLFY* expression pattern was largely complementary to the *MPL1* expression pattern (Fig. 5i, j).

### *MPL1* negatively regulates *CaLFY* expression

The *M. truncatula PALM1* is known to directly repress the *SGL1* expression, and *SGL1* upregulation induced the proliferation of extra leaflets in the *palm1* mutant[34]. We investigated the role of *CaLFY* in the proliferation of extra leaflets in *mpl1* mutant and found a significant *CaLFY* upregulation in both vegetative shoot apices and leaf primordia of the *mpl1-1* mutants when compared to WT (Fig. 6a, b). As the control, the *KNOXI* gene *STM* transcript level showed no significant difference between WT and *mpl1-1* (Supplementary Fig. 11d, e). In *mpl1-1* compound leaves, the number of LLs decreased in a gradient manner from proximal to distal, so we asked whether the genetic change is associated with a change in the pattern of *CaLFY* expression along the proximal-distal axis. Compared to WT leaf primordia where *CaLFY* has a higher expression level in distal portion than the proximal portion, the *mpl1-1* leaf primordia showed a nearly equivalent level of *CaLFY* expression in their distal and proximal portions (Fig. 6c), indicating that the upregulated *CaLFY* expression is more predominant in the proximal portion than the distal portion in *mpl1-1* leaf primordia.

RNA in situ hybridization was performed to detail the alterations in the pattern of *CaLFY* expression in *mpl1-1* mutant. In WT, *CaLFY* expression was abundant in the younger CLPs (P3 ~ P4) and exhibited a distinct gradient pattern, with stronger signals in the undifferentiated tip and weaker signals in the basal differentiated LL primordia; however, as differentiation became predominant at later stages (P5 and P6), *CaLFY* expression markedly decreased (Fig. 6d–i; Supplementary Fig. 13). Compared with WT, *mpl1-1* exhibited stronger hybridization signals of *CaLFY* in leaf primordia at different stages, particularly in

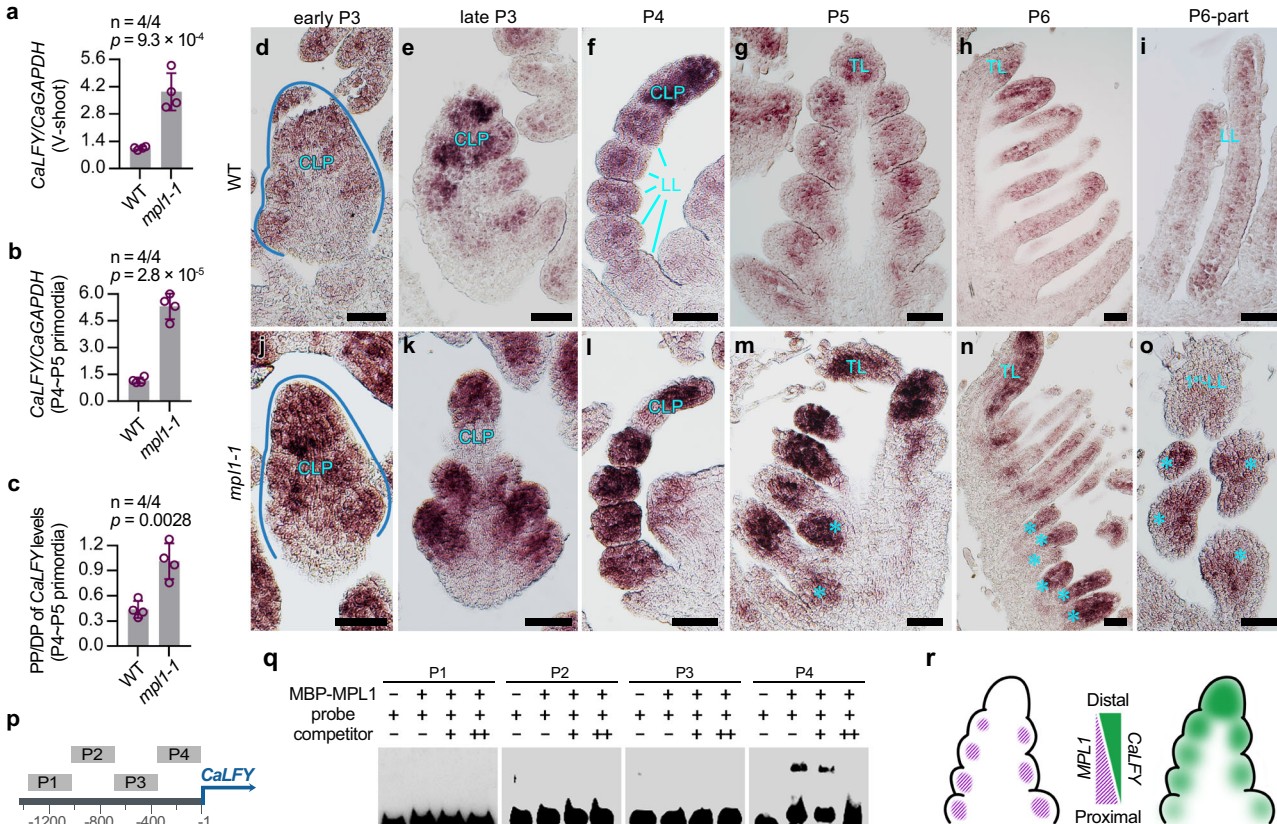

**Fig. 6 | *MPL1* negatively regulates *CaLFY* expression during the compound leaf development of chickpea.** RT–PCR analysis of *CaLFY* expression in vegetative-shoots (**a**) and leaf primordia (**b**) of WT and *mpl1-1*. **c** The ratio of *CaLFY* expression level in Proximal Portion (PP) to that in Pistal Portion (DP) of the P4–P5 leaf primordia of WT and *mpl1-1*. Data shows mean ± SD of 4 biological replicates in (**a**–**c**). The values above columns in (**a**–**c**) represents the *p*-value estimated by the two-sided unpaired Student's *t* test. **d–o** RNA in situ hybridization of *CaLFY* mRNA in leaf primordia of WT and *mpl1-1*. Shown are coronal sections through leaf primordia at the early P3 (**d**, **j**), late P3 (**e**, **k**), and P5 (**g**, **m**) stages, sagittal sections through leaf primordia at the P4 (**f**, **l**) and P6 (**h**, **n**) stages, and sections of parts of the P6 leaf primordia (**i**, **o**). Cyan asterisks indicate the 2nd-order lateral leaflet (LL) primordia in *mpl1-1*. Similar results were obtained from three independent experiments. Scale bars, 50 μm. **p** Diagram of the *CaLFY* promoter including P1-P4 sequences used for electrophoretic mobility shift assay (EMSA). **q** EMSA analysis of P1-P4 sequences labeled with biotin in the presence and absence of purified recombinant MBP–MPL1 proteins. Competition binding was performed in the presence of 10-fold (+) or 50-fold (++) unlabeled probes. Similar results were obtained from three independent experiments. **r** Model for *MPL1* and *CaLFY* expression patterns and their interaction in WT P4 leaf primordium. Source data for Figs. 6a–q are provided as a Source Data file.

their proximal regions. From stages P3 to P4, high levels of *CaLFY* mRNA were detected not only in the undifferentiated tip of the CLP but also in the proximal first-order LL primordia (Fig. 6j–l; Supplementary Fig. 14a–c). During later stages (P5 ~ P6) when the initiation of second-order LL primordia became prominent, the first-order LL primordia retained robust *CaLFY* expression, and the second-order LL primordia also showed strong hybridization signals (Fig. 6m–o; Supplementary Fig. 14e, f). We therefore concluded that the *mpl1* mutation induced *CaLFY* upregulation more dramatically in the proximal portion than the distal portion of the early CLP, disrupting the *CaLFY* expression gradient along the longitudinal axis.

Given that *MPL1* functions as a transcription factor, electrophoretic mobility shift assays (EMSAs) were performed to examined if MPL1 is capable of binding to the promoter region of *CaLFY*. The results show that the MBP-MPL1 fusion protein can bind to the promoter of *CaLFY* between −1 and −360 bp (P1), but not to the region between −361 and −1440 bp (Fig. 6p, q). This indicates that *MPL1* suppresses *CaLFY* expression by directly binding to its promoter. In conclusion, we uncovered that *MPL1* and *CaLFY* are arrayed in complementary expression gradients along the proximal-distal axis of early leaf primordium functioning as a morphogenic gradient, providing a key mechanism for the sequential pattern of leaflet development (Fig. 6r).

## MPL1 integrated with the auxin signaling pathway to regulate the compound leaf development of chickpea

To further elucidate the potential mechanism of *MPL1*-mediated regulation of compound leaf development, transcriptome analysis was carried out using RNA-seq on samples of the vegetative shoot apices containing SAM and P1 ~ P6 primordia (Supplementary Data 2). Compared to the WT, a total of 1002 genes were up-regulated and 656 genes were down-regulated in *mpl1-1* (Supplementary Fig. 13a, b; Supplementary Data 3, 4). Many auxin related genes were significantly differentially expressed (Supplementary Data 5). Several members of the *Auxin-Response Factor* (*ARF*) family were enriched in the upregulated set (Fig. 7a). Therein, *CaARF11* is highly homologous to the tomato *SlMP*, which is known to promote leaflet initiation and outgrowth;[16] three putative cytokinin dehydrogenase, as well as *CaCUC3* and *CaFCL1*, whose homologs have been shown to regulate the boundary formation between leaflets, were also altered in their expression in *mpl1* (Fig. 7a). These differential expression results were confirmed by RT-qPCR analysis on samples consisting dissected leaf primordia (P4 ~ P6) (Fig. 7b, Supplementary Fig. 15c). In these samples, an *ARF* gene, *CaARF23*, was increased approximately two hundred-fold in *mpl1-1* (Fig. 7b). By RNA in situ hybridization, *CaARF23* expression was found to be low in WT leaflet primordia, but significantly upregulated in the first-order LL primordia of *mpl1-1* leaves at an early

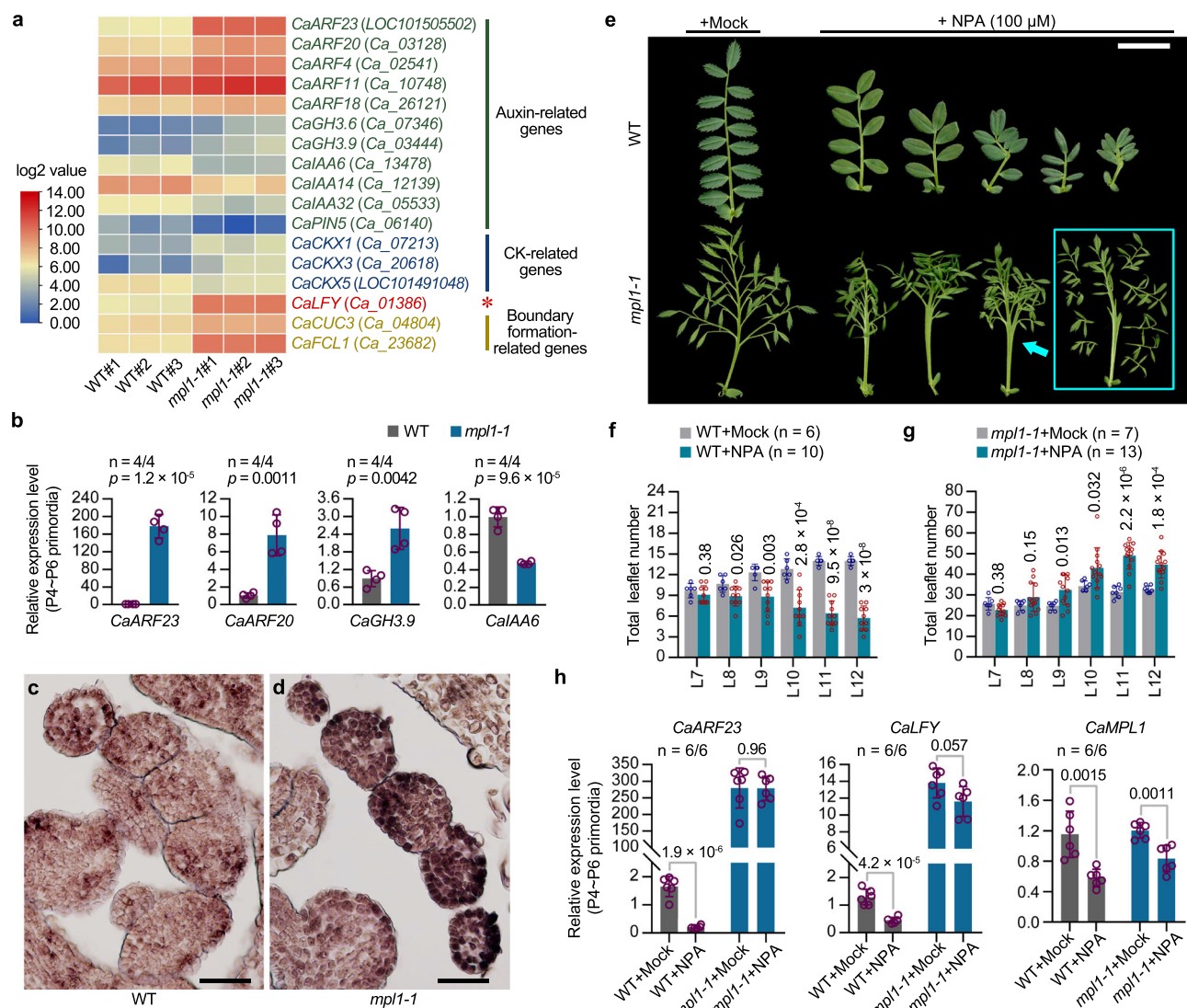

**Fig. 7 | Transcriptional control of the auxin signaling pathway during compound leaf patterning by *MPL1*. a** Gene expression differences between WT and *mpl1-1* shoots using RNA-seq transcriptome analysis. A heat map was created using fragments per kilobase of exon model per million mapped reads (FPKM). WT and *mpl1-1* biological replicates were plotted in the left three columns and right three columns, respectively. **b** RT–qPCR validation of expression differences of certain selected genes between WT and *mpl1-1*. *CaGAPDH* was used as an internal reference gene. The samples used for the analysis consist of a mixture of P4 - P6 leaf primordia dissected from the shoot apex. Data shows mean ± SD of 4 biological replicates. The values above each column represent the *p*-values estimated by the two-sided unpaired Student's *t* test. **c, d** RNA in situ hybridization of *CaARF23* mRNA in P4 leaf primordia of WT and *mpl1-1*. Similar results were obtained from three independent experiments. Scale bars, 50 μm. **e** Compound leaf development responses to the auxin transport inhibitor N–1-naphthylphthalamic acid (NPA). Two-week-old plants of WT and *mpl1-1* grown in soil were sprayed by mock or 100 μM NPA once every three day, for a duration of 15 days; compound leaves from

node 11 (the third internode beneath the shoot apex) were photographed at three weeks after the spraying stopped. Scale bar, 2 cm. Quantification of total leaflet number in compound leaves from node 7 to node 12 of 7-week-old WT (**f**) and *mpl1-1* (**g**) plants after spraying with mock (indicated as "+Mock") or 100 μM NPA solutions (indicated as "+NPA"). Data shows mean ± SD (*n* = 6 "WT+Mock" plants, 10 "WT + NPA" plants, 7 "*mpl1−1*+Mock" plants, and 13 "*mpl1-1* + NPA" plants). The values above bars represent the *p*-values of the two-sided unpaired Student's *t* test between the "+Mock" and "+NPA" groups. **h** RT–PCR analysis of *CaARF23* (left), *CaLFY* (middle) and *MPL1* (right) expression levels in leaf primordia of WT and *mpl1-1* after spraying with mock or 100 μM NPA solutions. *CaGAPDH* was used as an internal reference gene. The samples consist of a mixture of P4 to P6 leaf primordia. Data shows mean ± SD of 6 biological replicates. The values above columns represent the *p*-values estimated by the two-sided unpaired Student's *t* test between the "+Mock" and "+NPA" groups. Source data for Fig. 7b, f, g, h are provided as a Source Data file.

developmental stage (Fig. 7c, d). These results suggested that *MPL1* also regulates several critical aspects of leaf development, among which the auxin signaling pathway is particularly important.

We therefore investigated the effect of auxin-transport inhibitor 1-N-naphthylphthalamic acid (NPA) on the leaf development (Supplementary Fig. 14a). NPA-treatment resulted in smoother leaflet margins and reduced growth in the length of compound leaves and leaflets (Fig. 7e, Supplementary Fig. 16b). Consistent with observations in many species[9,18,39–41], the treatment inhibited the leaflet formation in

WT plants (Fig. 7e, f), accompanied by a significant downregulation of *CaARF23* and *CaLFY* (Fig. 7h). However, it potently increased the leaflet production in *mpl1-1* (Fig. 7e, g). This is perhaps attributed to multiple causes, including the unaffected extremely-high expression of *CaARF23* and *CaLFY* by the treatment (Fig. 7h), an important role of auxin transport in promoting the differentiation process[42,43], and possible mechanistic links between *LFY*, *ARF* and the auxin signaling pathway[39,44]. As a control, *MPL1* expression is not changed by its own mutation, and is slightly down-regulated by the treatment in both WT

and *mpl1* (Fig. 7h). These observations suggest that the MPL1-CaLFY module is integrated with the auxin signaling pathway to regulate compound leaf morphogenesis.

## Discussion

In this study, we cloned the naturally occurring mutant *multi-pinnate leaf* (*mpl1*) of chickpea that has been known in the literature for more than 60 years[31,32]. Through BSA-Seq, linkage analysis, and genotyping of multiple independent alleles, we identified *Ca_02268* as the gene responsible for the *mpl1* leaf phenotype. Heterologous complementation of the *M. truncatula palm1* mutant indicated that *Ca_02268* is the ortholog of *PALM1*. Moreover, its expression pattern is consistent with the observed *mpl1* phenotype. Based on these results as well as its molecular association with *CaLFY*, we conclude that *Ca_02268* is definitely the *MPL1* gene.

Compound leaf development has long fascinated developmental biologists because it provides insight into the trade off between differentiation and the maintenance of undifferentiated tissues with the potential to generate new structures. In the case of chickpea pinnate leaves, leaflet initiation follows an acropetal pattern that involves a morphogenetic gradient along the proximodistal axis of early leaf primordia, with young undifferentiated cells at the tip and more differentiated cells at the base (Supplementary Fig. 17a–c). Our study suggests that this gradient is related to the molecular interaction between *MPL1* and *CaLFY*. *CaLFY* functions to maintain the undifferentiated status while *MPL1* promotes differentiation through directly repressing *CaLFY* expression. During the process of LL initiation (P2 - P4 stages), *CaLFY* expression is high in the undifferentiated tip of the whole CLP but lower in the more differentiated LL primordia (Supplementary Fig. 17b, c). In contrast, *MPL1* expression is seen throughout each differentiated LL primordium, but barely detected in the undifferentiated tip (Supplementary Fig. 17b, c). *MPL1* expression shows a spatio-temporal pattern that starts from the most basal LL primordia, progresses acropetally as LL primordia sequentially formed, and eventually extends to the differentiated TL primordium (Supplementary Fig. 17b–d). As the LL primordia are basipetally increased in size (Fig. 1c–f), it is reasonable that the more differentiated LL primordia consistently have an earlier and broader expression of *MPL1* than younger ones. In the *mpl1* mutant, the LL primordia have a high expression of *CaLFY*, and thus were converted as "pseudo undifferentiated tips" (Supplementary Fig. 17e–i). We considered that this conversion follows a developmental gradient manner. In detail, the most basal LL primordia were converted as the early undifferentiated tip, which has stronger capability to generate new leaflet primordia, whereas the more distal LL primordia were converted as the late undifferentiated tip, which soon later was differentiated into the TL primordium and thus has less potential to generate new primordia (Supplementary Fig. 17f, g, i); consequently, the more proximal first-order leaflets can produce more higher-ordered leaflets.

This study together with previous works indicate that the *MPL1-CaLFY* of chickpea and *PALM1-SGL1* of *M. truncatula* are evolutionarily-conserved modules, but exhibit different developmental outputs during compound leaf morphogenesis[24,34,37]. Chickpea leaf development has a long period of leaflet initiation, accompanied by a sustained, high, and broad expression of *CaLFY* (Fig. 6d–g). However, trifoliolate leaf development in *M. truncatula* involves only one leaflet initiation event; a transient and restricted strong expression of *SGL1* indicated a short period of undifferentiated status in the tip of the early CLP (Supplementary Fig. 18a)[23,24,37]. In the *palm1* mutant, the earliest two LL primordia (P2) acquire a temporary elevated expression of *SGL1* and thus were converted as "pseudo undifferentiated tips"; they therefore possess a limited potential to initiate one new leaflet primordium at the later stage (P3) (Supplementary Fig. 18b). In fact, in this mutant, when all five leaflet primordia are formed, each LL primordium was further comparable to "a pseudo differentiated TL primordium",

with an expression pattern of *SGL1* similar to that of the TL primordium[23]. This is further evident when the *palm1* mutant was combined with the *pinna1* mutant that disrupts the gene encoding a BEL1-like homeodomain protein (Supplementary Fig. 18c)[23]. In the *palm1 pinna1* double mutant, the first-order LLs were converted as "pseudo TLs" by the *palm1* single mutation, and thus additional "second-order LLs" were formed due to the *pinna1* mutation (Supplementary Fig. 18d). Therefore, *MPL1* and *PALM1* function as key regulators to endow the differentiated identity to the early LL primordia, but it cannot be disregarded other regulators, such as PINNA1, in controlling a precise spatiotemporal expression of *CaLFY/SGL1* to maintain a robust differentiation program for leaflet primordia.

In many compound-leafed species, *KNOXI* genes are activated in early leaf primordia and function to maintain the undifferentiated state[19,45]. In tomato and *Cardamine hirsute*, loss-of-function of these genes resulted in simple leaves, and their overexpression led to increased leaflet production[8,46–48]. However, it is now recognized that *LFY* genes are also widely required for this function. In the fern *Ceratopteris richardii*, both *CrKNOXI* and *CrLFY* genes were expressed in the complex frond primordia of the young sporophyte, and suppression of *CrLFY* activity by RNAi resulted in simple fronds, suggesting an ancestral function of *LFY* in maintaining the undifferentiated tissues of vegetative organs[49,50]. In most legumes (outside of the IRLC), *KNOXI* genes were expressed in their compound leaves, but *LFY* orthologs also have important roles[19]. In *L. japonicus* and soybean, the loss-of-function mutation or RNAi silencing resulted in reduced leaflet number[19,27]. In mung bean, the loss-of-function mutant of the *LFY* gene *UNIFOLIATE LEAF* (*UN*) resulted a completely simple leaf phenotype[26]. In IRLC legumes, as previously reported, *KNOXI* genes are not associated with the compound leaf development and *LFY* genes completely take place the *KNOXI* function. Our study strengthens this idea and meanwhile suggests that *LFY* activity is subjected to delicate spatiotemporal regulation thus to form a specific compound leaf pattern. However, recent works in *M. truncatula* revealed that *KNOXI* can complement the *sgl1* mutant and their overexpression dramatically enhances leaf complexity[51], but overexpression of *SGL1* (*35 S::SGL1*) has no effect on the compound leaf pattern[38]. Moreover, in *C. hirsuta*, loss or gain of *LFY* function only affects the progression of leaf heteroblasty[52]. These indicate that *LFY* acts to specify an undifferentiated state in a context-dependent manner, requiring interaction with other factors, different from *KNOXI* which maintains an undifferentiated state despite developmental context. Future studies to dissect the LFY interacting partners in legumes, such as STAMINA PISTILLOIDA (STP) of pea, which is orthologous to the F-box protein UNUSUAL FLOWER (UFO) and leads to leaflet reduction after mutation[53], will provide new insight into the LFY regulatory architecture maintaining the undifferentiated state.

Another significant feature of *mpl1* leaves is the greatly reduced leaflet size (Fig. 2f, g), which is also found in leaves of the *palm1 pinna1* double mutant[23], indicating an antagonistic interplay between maintaining morphogenesis and promoting expansion. It seems to be a clever strategy of compound-leafed plant that an increase in the number of leaflets was accompanied with decrease in leaflet area, which can avoid the occlusion between the leaflets and ensure the efficiency of photosynthesis and ventilation. This is in agreement with the "compensation" phenomenon, such as many plant organs decrease in size or weight when they increase in number[54]. *MPL1* was highly expressed during leaf maturation (from P5 - P8) (Fig. 4b, c), consistent with its possible role in promoting leaflet outgrowth. Cellular imaging analysis suggested that the small leaflets in *mpl1* are mainly caused by reduced cell proliferation (Fig. 2h–j). We found that the expression levels of *CaCKX3* and *CaCKX5* were considerably higher in *mpl1* than WT (Fig. 7a, Supplementary Fig. 15c). A recent study revealed that *NsCKX3* of *Nicotiana sylvestris* plays a pivotal role in leaf blade expansion; overexpression of *NsCKX3* resulted in narrower and shorter leaves, while its downregulation led to wider leaves[55]. Therefore *MPL1*

may contribute to cytokinin homeostasis through regulating CKXs during blade outgrowth. Several closely related C2H2 homologs play significant roles in plant organ development by repressing genes involved in cell division. *Aquilegia POP* regulates spur development by promoting cell division in the spur cup[36]. *A. thaliana RBE* and pea *STIPULES REDUCED* (*St*) were in a sister relationship with *PALM1/POP/MPL1* (Supplementary Fig. 8). *RBE* promotes cell proliferation at the boundaries of petal primordia by controlling the cell fate transition from mitotic growth to differentiation, while *St* regulates both cell division and cell expansion involved in stipule development[56,57]. The narrow petal of the *rbe* mutant and the smaller stipule of the *st* mutant were reminiscent of the small leaflets in *mpl1*[56,57]. *A. thaliana SUP* and rice *SMALL REPRODUCTIVE ORGANS* (*SRO*) formed another clade sister to *PALM1/POP/MPL1* and *RES/St* (Supplementary Fig. 8), both of which affect cell division in flower development[58,59]. Thus, these findings suggested that MPL1 homologs have similar effects in organ morphogenesis, specifically in regulating cell division. Therefore, *MPL1* not only plays an essential role in promoting a differentiated fate of the leaflet primordia through repressing the *CaLFY* expression, but may also have a significant function in promoting leaflet blade expansion through regulating cell division.

## Methods

### Plant material and growth conditions

PI587041/*mpl1-1* (Desi type), PI587039/*mpl1-2* (Desi type), PI587040/*mpl1-3* (Kabuli type), ICCV96029 (Desi type) and CDC Frontier (Kabuli type) were obtained from the United States Department of Agriculture (USDA) Western Regional Plant Introduction Station at Pullman, WA, provided by Dr. Clarice Coyne. ICCV96029 and CDC Frontier were used as the wild type (WT). The *M. truncatula palm1-5* mutant were isolated from a *Tnt1* mutant collection (https://medicago-mutant.dasnr.okstate.edu/mutant/)[23,34], and the cultivar (cv.) R108 was used as the WT. The plant materials used in the RNA in situ hybridization, bulked segregant analysis (BSA), RNA sequencing (RNA-seq), SEM, and N-1-naphthylphthalamic acid (NPA) treatment assays were grown in greenhouses under the following controlled conditions: a relative humidity of 50–60%, a temperature range of 24 °C during the day and 20 °C at night, a light intensity of 150 μmol/m²/s, and a 16-h light/8-hour dark cycle. For seed propagation, the plants were planted in the experimental farm located at Xishuangbanna Tropical Botanical Garden in Yunnan, China.

### Scanning electron microscopy (SEM)

Shoot apices from 2- to 4-week-old plants were fixed in a fixation solution (10% formaldehyde, 5% acetic acid and 50% ethanol) for vacuum infiltration 30 min and incubated at room temperature overnight. Plant tissues were further dehydrated in a graded ethanol series (60%,70%, 80%, 90%, 95% and 100% ethanol for 30 min each) and dried in liquid $CO_2$ with a critical-point drier (Samdri-PVT-3D, Tousimis, USA). After dissected under a stereomicroscope (SZX16, Olympus) and coated with gold, tissue samples were then examined under a SIGMA 300 SEM (Zeiss, Germany) at an accelerating voltage of 5 kV.

### Plasmid construction and generation of transgenic plants

To construct the *PALM1pro::cMPL1-NOS-T* plasmid, the full-length coding sequence of *MPL1* was amplified from a chickpea WT (cv. CDC Frontier) vegetative shoot cDNA sample, while a 5 kb promoter fragment of *PALM1* was amplified from a *M. truncatula* WT (cv. R108) leaf gDNA sample. The two fragments were subsequently assembled into the pCAMBIA3301 vector using a ClonExpress II One Step Cloning Kit (C112, Vazyme, China). Sanger sequencing was performed to confirm the integrity of the construct. The primer sequences are listed in Supplementary Data 6. For complementation analysis, the *PALM1-pro::cMPL1-NOS-T* plasmid was introduced into *Agrobacterium tumefaciens* strain EHA105 and used for transformation of *palm1-5* via *Agrobacterium*-mediated transformation[60].

### Subcellular localization

To construct the *35 S::GFP-MPL1*, the *GFP*-coding sequence and the full-length *MPL1*-coding sequence were assembled into the pCAMBIA3301 vector using the ClonExpress II One Step Cloning Kit (C112, Vazyme, China). The construct was transiently expressed in *Nicotiana benthamiana* leaves via *Agrobacterium*-mediated infiltration. GFP signal was observed after 36 h of dark incubation at 22 °C using a confocal laser scanning microscope (FV1000, Olympus, Japan) with excitation at 488 nm.

### RNA extraction, RT–PCR and RT–qPCR

Total RNA was extracted from various tissues and leaf primordia at different developmental stages using a RNAsimple Total RNA Kit (DP419, Tiangen, China) according to the manufacturer's instructions. Subsequently, 2 μg of total RNA was utilized to synthesize first-strand cDNA with the HiScript® II 1st Strand cDNA Synthesis Kit (R212, Vazyme, China). For the reverse transcription PCR (RT-PCR) assay in *M. truncatula*, the EasyTaq enzyme (AP111, TransGen, China) was employed with *MtACTIN* serving as a control. Reverse transcription quantitative PCR (RT-qPCR) assays in chickpea were conducted using TransStart Tip Green qPCR SuperMix (AQ141, TransGen, China) on the Roche Light-Cycler480II instrument, with *CaGAPDH* used as an internal reference gene. To determine expression levels, at least three biological replicates were performed, each with independent RNA isolations and three technical repeats. Fold changes were calculated from the $2^{-\Delta\Delta Ct}$ values. The primer sequences are listed in Supplementary Data 6.

### RNA in situ hybridization

Vegetative shoot apices from 2-week-old WT (cv. CDC Frontier) and *mpl1-1* plants were fixed in a fixation solution (10% formaldehyde, 5% acetic acid and 50% ethanol), then transferred into embedding cassettes and fixed overnight using an automated tissue processor (Leica ASP200S, Wetzlar, Germany). Next, the samples were embedded in paraffin by HistoCore 86 Arcadia (Leica, Germany).

To prepare the RNA probes, the coding sequence (CDS) regions of *MPL1,CaLEAFY, CaBP1, CaSTMa* and *CaARF23* were cloned into pEASY-Blunt cloning vector (CB101, TransGen, China). The resulting pEASY-target plasmids were served as templates for PCR and amplified with specific primers of which the reverse primer contained a T7 RNA polymerase promoter (5′-TGTAATACGACTCACTATAGGGC-3′) at its 5′ terminus. The further resulting PCR products were used as templates to transcribe RNA digoxigenin-labeled probes in the presence of T7 RNA polymerase (10881767001, Roche, Switzerland), 10×DIG RNA Labeling Mix solution (11277073910, Roche, Switzerland) and RNAase inhibitior (3335399001, Roche, Switzerland). These probes were then hydrolysed to an average length of 150–200 bp. The vegetative-shoot apices were sectioned into 8-μm-thick sections using a Leica RM 225587 microtome (Leica, Germany). Hybridization, washing and staining were carried out as described[61], with minor modifications. In brief, following dehydration and rehydration in a gradient concentration alcohol solution, the sections were digested in protease K solution buffer for 28 min. The sections were then hybridized with corresponding probes. Blotting was performed with Anti-digoxigenin AP-conjugate (11093274910, Roche, Switzerland) for 2 h, followed by incubation with the NBT solution (11383213001, Roche, Switzerland) for 24 h. Optical photographs of the sections were captured with an Olympus BX63 microscope. To better analysis the expression level and pattern of *CaLEAFY* and *CaARF23* between WT and *mpl1-1*, the paraffin sections of WT and *mpl1-1* were placed on the same slide for hybridization.

### Bulk segregation analysis

To identify the causal gene responsible for the *mpl1-1* mutation, we performed a cross between WT (ICCV 96029) and the *mpl1-1* mutant (PI587041) as reported[62] and generated an F2 population. The WT and *mpl1-1* DNA mix pools were prepared by mixing equal amounts of

genomic DNAs from 28 WT F2 individuals and 28 *mpl1-1* F2 individuals, respectively. The two DNA mix pools and two parental DNA samples (the female parent ICCV 96029 and the male parent PI587041) were extracted using a EasyPure Plant Genomic DNA Kit (EE111, TransGen, China). Subsequently, we sequenced the DNA samples using the Illumina HiSeq4000 platform (Novogene, Beijing, China).

To ensure reliability and eliminate artificial bias in the reads for downstream analyses, we processed the raw data (raw reads) through several quality control (QC) procedures using the Fastp software[63]. Next, we aligned the clean reads of each sample to the reference genome using BWA (Burrows-Wheeler Aligner)[64]. We performed variant calling for all samples using the Unified Genotyper function in GATK3.8[65], and annotated SNP or InDel based on the GFF3 files of the chickpea reference genome (CDC Frontier v1.0) using ANNOVAR[66,67].

To calculate the SNP/InDel index, we obtained read depth information for homozygous SNPs/InDels above in the two extreme pools. We used a window size of 1 Mb and a step size of 10 kb as default settings to average all SNP/InDel indexes in each window, which was then used as the SNP/InDel index for that window. We calculated the difference of the SNP/InDel index between the two pools as the delta SNP/InDel (ΔSNP/InDel) index[68].

To narrow down the list of candidate genes in this region, we filtered the SNPs and InDels by simultaneously satisfying the following criteria: (1) deleterious variant in the exon that greatly influenced the function of protein; (2) homozygous mutation (aa) in both mutant mix pool and PI587041 parent sample, heterozygous mutation (Aa) in WT mix pool, and WT genotype (AA) in ICCV 96029 parent sample. After applying these filters, we identified five genes with genomic variants that met all these criteria in the candidate region, which are listed in Supplementary Data 1.

### Genomic PCR sequencing and linkage analysis

Full length genomic sequences of *MPL1* from individuals of WT, *mpl1-1*, *mpl1-2* and *mpl1-3* were verified by PCR-based sequencing with the primers listed in Supplementary Data 6. The PCR reaction was performed under the following cycling conditions: 95 °C for 2 min; 94 °C for 20 s, 58 °C for 20 s, 72 °C for 2–5 min, 32 cycles; 72 °C for 5 min. To test whether the *mpl1* phenotype is link to the mutation in *Ca_02268*, a linkage analysis was conducted. Given the *mpl1-1* mutant carries a specific 14 bp deletion that removes a *Bst*C8I recognition site in *Ca_02268*, we amplified the full length genomic sequences of *MPL1* from various samples: WT (CDC frontier), a *mpl1-1* mutant, a $BC_2F_1$ plant and 45 $BC_2F_2$ individuals exhibiting the *mpl1-1* mutant phenotype. Subsequently, we digested the PCR products with *Bst*C8I restriction enzyme (R0579L, New England Biolabs, USA) at 37 °C for 2 h and then fractionated the resulting fragments by 1% agarose gel electrophoresis. This allowed us to determine if the *mpl1* phenotype was indeed related to the mutation in *Ca_02268*, as any changes in the restriction fragment pattern would be indicative of linkage.

### RNA-seq analysis

Total RNA was extracted from the vegetative shoots of 2-week-old WT (CDC Frontier) and *mpl1-1* plants using the RNeasy Plant Mini Kit (74904, QIAGEN, Germany). For each genotype, a total of 25 vegetative shoots from 25 individuals were pooled to constitute one biological replicate. Three biological replicates were prepared for each genotype. The integrity of RNA was evaluated by using the RNA Nano 6000 Assay Kit of the Bioanalyzer 2100 system (Agilent Technologies, CA, USA). RNA-Seq libraries were constructed with the NEBNext® Ultra™ Directional RNA Library Prep kit (E7420, New England Biolabs, USA) and subsequently sequenced via an Illumina Novaseq 6000 platform (Novogene, Beijing, China). Raw data were filtered using Fastp, and the resulting clean reads were then mapped onto the chickpea reference genome (CDC Frontier GA v1.0) using Hisat2 (v2.0.5). FeatureCounts (v1.5.0-p3) was utilized to count the read numbers mapped to each gene, and subsequently, FPKM for each gene was calculated based on the length of the gene and the number of reads mapped to it. Differential expression analysis between the two genotypes was performed utilizing the DESeq2 R package (1.20.0). The resulting *P*-values were adjusted using the Benjamini and Hochberg approach to control the false discovery rate. The thresholds for determining significant differential expression were set as padj <= 0.05 and |log2(fold change)| >= 1. The creation of a heatmap was facilitated by Helm software (Heatmap Illustrator, version 1.0) and Tbtools[69].

### N-1-naphthylphthalamic acid (NPA) treatment

The NPA was dissolved in DMSO to form a 100 mM stock solution, and then diluted into 100 μM with water for treatment. Two-week-old plants of WT and *mpl1-1* grown in soil were sprayed by mock (0.1% DMSO) or 100 μM NPA once every three day for 15 days; the whole plants and compound leaves were photographed with a Nikon D7100 camera at 3 weeks after the spraying stopped.

### EMSA

The *MPL1* coding sequence was cloned into the protein expression vector pMAL-c2x by the ClonExpress II One Step Cloning Kit (C112, Vazyme, China). The MBP-MPL1 fusion protein was expressed in *E.coli* Rossetta (DE3) by induction with 0.5 mM isopropyl β-D-thiogalactopyranoside (IPTG, A100487, Sangon Biotech, China) overnight at 16 °C in an orbital shaker for 6–10 h. The bacteria were then harvested, washed, resuspended, and lysed. The MBP-MPL1 protein was purified from the crude extract by amylose resin affinity chromatography (E8201S, New England Biolabs, USA) following the manufacturer's protocol.

For the EMSAs, the probes were prepared by amplifying PCR products from the *MPL1* promoter using Primestar HS DNA Polymerase (R010Q, Takara, Japan), with primers that were 5'-end labeled with biotin. The same primers without labeling biotin were used for the amplification of PCR products to act as competitors. All PCR products were purified using the Easypure Quick Gel Extraction Kit (EG101, TransGen, China).

To conduct the binding reaction, 20 fmol biotin-labeled probes and 200 ng purified MBP-MPL1 proteins were incubated with the binding buffer at 18 °C for 20 min. In some reactions, the unlabeled probes were added for competition. The complex was then resolved on 5% polyacrylamide gels and transferred onto nylon membranes using the Trans-Blot Turbo Blotting Transfer System (1704150, Bio-Rad, USA). Using the Chemiluminescent EMSA kit (GS009, Beyotime Biotechnology, China), the membrane was blocked, then probed with a streptavidin-HRP conjugate, and finally, the DNA-protein complex was visualized by chemiluminescence imaging (Tanon 5200 Multi, Tanon Biomart, China).

### Phylogenetic analysis

Phylogenetic analysis was conducted by retrieving MPL1 homologs from Phytozome (https://phytozome.jgi.doe.gov/pz/portal.html) for most species. For some species, homologs were identified through BLASTp searches in the following specific databases: the *Buxus sinica* and *Tetracentron sinense* (https://doi.org/10.5061/dryad.cjsxksn6d), the gymnosperms *Picea abies* (http://congenie.org/) and *Ginkgo biloba* (http://gigadb.org/), the legumes *L. japonicas* (http://www.kazusa.or.jp/lotus/) and pea (https://urgi.versailles.inra.fr/Species/Pisum), the *Musa* (https://banana-genome-hub.southgreen.fr/) and the *Cardamine hirsuta* (http://chi.mpipz.mpg.de/blast.html). To ensure accuracy, sequences from each species were further checked by carrying out a multiple sequence alignment and phylogenetic tree estimation. All valid MPL1 homologs were presented in Supplementary Data 7.

Multiple alignments of MPL1 homologs were generated using ClustalX (v2.1) with default parameters. The maximum likelihood method was used to perform the phylogenetic tree analysis with

IQTREE v1.6.10 using the JTT + F + G4 model suggested by the IQTREE model test tool (BIC criterion)[70]. Ultrafast bootstrap replicates of 2000 and iterations of 5000 were used to ensure statistical significance of the results. The tree was then edited using the MEGA 5.0 program and manually optimized for viewing clarity. The fully phylogenetic tree is output as an NWK file (Supplementary Data 8).

## Statistics and reproducibility
The width, length and area of leaflets, as well as the area of epidermal cells, were quantified by ImageJ. Throughout, error bars shown represent the standard deviation (SD) of the mean for all numerical values. Statistical tests were performed using GraphPad Prism (version 8.0.2) and involved unpaired two-sided Student's $t$-test. Significant differences in all multiple comparisons were denoted by different letters at $p < 0.05$, according to the $t$-test. The significance of pairwise comparisons was directly presented as the corresponding $p$-values. No statistical methods were used to pre-determine sample sizes. Experiments were not randomized. The investigators were not blinded to allocation during experiments and outcome assessment.

## Reporting summary
Further information on research design is available in the Nature Portfolio Reporting Summary linked to this article.

## Data availability
All raw reads, including the BSA sequencing data and the RNA-seq data, are accessible through NCBI BioProject accession number PRJNA984229. The source data for Figs. 2, 4–6, 8 and Supplementary Figs. 6, 7, 9, 11, 15 are provided as a Source Data file. All other data that support the findings of this study are available at https://doi.org/10.6084/m9.figshare.22985897, or can be obtained from the corresponding author upon request. Requests for plant materials should be addressed to L.H.. Source data are provided with this paper.

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

## Acknowledgements

We thank other members of the J.C. laboratory for their valuable inputs, Drs. Clarice Coyne (USDA-ARS Plant Germplasm Station, Pullman, WA) for providing the chickpea seeds and Douglas R. Cook (University of California, Davis, CA, USA) for information on the chickpea mini core collection, and Kuanqiang Tang (Northeast Institute of Geography and Agroecology, Chinese Academy of Sciences) for providing the computer servers. We also thank the Institutional Center for Shared Technologies and Facilities of Xishuangbanna Tropical Botanical Garden, CAS for providing the computer resources and technical support. Research in J.C. laboratory was supported by Strategic Priority Research Programs of the Chinese Academy of Sciences (Grants No. XDA26030301 and XDB27030106), National Natural Science Foundation of China grants (32070204, U2102222, 32170839, and 32170360), CAS-Western Light 'Cross-Team Project-Key Laboratory Cooperative Research Project' (xbzg-zdsys-202016), Yunnan Revitalization Talent Support Program (Grant Nos: XDYC-QNRC-79, XDYC-QNRC-2022-0179 and XDYC-QNRC-2022-0335), Yunnan Fundamental Research Project (Grant 202101AW070004), and Youth Innovation Promotion Association CAS (Grant 2021395). Research in M.T. laboratory was supported by Agriculture and Food Research

Initiative Grant No. 2015-67014-22888 from the USDA National Institute of Food and Agriculture.

## Author contributions

Y.L., J.C and L.H. designed research; Y.Y., Y.L., X.J., M.L., Y.H., W.L., X.M. and Z.G. performed research; R.W., B.Z., B.P., and X.Z. analysis data; and Y.L., M.T., J.C. and L.H. wrote the paper.

## Competing interests

The authors declare no competing interests.
