## [Peer Review File · Nature Communications]

Control of compound leaf patterning by MULTI-PINNATE
LEAF1 (MPL1) in chickpeaREVIEWER COMMENTS

Reviewer #1 (Remarks to the Author):

The development of compound leaves has long fascinated developmental biologists because it informs our fundamental understanding of the trade off between differentiation and the maintenance of undifferentiated cells with the potential to generate new structures. In the manuscript entitled "Control of compound leaf patterning by MULTI-PINNATE LEAF1 (MPL1) in chickpea" Liu et al. beautifully reveal how a gradient between CaLFY maintaining undifferentiated status from the tip downward and MPL1 promoting differentiation from the base upward controls the initiation of leaflets. They first show stunning SEM images which reveal the acropetal (base to tip) development of leaflets in the wild type chickpea leaf and how this pattern is enhanced in *mpl1* mutants where leaflet form on the lower leaflets. The acropetal development of leaflets is interesting because it is the reverse of other well studied compound leaf systems such as tomato and Cardamine in which leaflets develop basipetally. The authors positionally clone the MPL1 gene and find it encodes a C2H2 zinc finger transcription factor ortholog of PALM1, which is known to regulate leaflet number in *Medicago*. With beautiful in situ hybridization images, the authors show that MPL1 is expressed in developing leaflets starting from the base and moving upward as the leaflets mature. In basipetal leaflet systems, the expression of KNOX genes (STM and BP) promotes the maintenance of the undifferentiated state. Similar to pea, the authors show that CaSTM and CaBP1 are expressed only in the meristem, so they are not candidate factors. Again similar to Pea they find that CaLFY is expressed in developing leaves, particularly in the undifferentiated tip and youngest leaflets. CaLFY is strongly upregulated in *mpl1* mutants, and the authors show that MPL1 can bind to the CaLFY promoter using EMSAs, suggesting that MPL1 directly represses CaLFY expression, this promoting differentiation. Finally the authors examine auxin transport because auxin transport has been shown to promote leaflet and serration formation in other studies. The author's treatment of wild type with auxin transport inhibitor NPA shows that auxin transport does promote leaflet and serration formation. However, the interaction with *mpl1* is complex because NPA increases leaflet formation in the mutant. CaARF23 which is normally downregulated in NPA treated wild type plants is not downregulated in NPA treated *mpl1* mutants. I suggest the authors use caution in interpreting the NPA results. Overall, the manuscript is well written, the figures are beautiful, and the experiments are well done with appropriate controls. I think this manuscript adds an important piece to our understanding of compound leaf development and the different ways plants balance differentiation and maintaining undifferentiated tissues.

Minor comments:

"(ii) auxin polar transport is connected to the mechanism of MPL1 function in controlling the spatio-temporal expression of CaARF23 and CaLFY during leaf development." The data do not support this interpretation. I believe here the authors should use caution and just point to the complex interaction. It will take another whole paper worth of experiments to understand this interaction better, which is for the future.

"We found that the NPA-treatment significantly down-regulated the CaARF23 and CaLFY expression in WT, but did not affect the extremely high expression levels of these genes in *mpl1-1* (Fig. 7f)." The figure referenced should be 7h.

"By contrast, the *mpl1* mutation did not change its own expression, and the NPA-treatment slightly down-regulated the MPL1 expression in both WT and *mpl1* (Fig. 7f)." Again the figure

referenced should be 7h.

“Although we did not perform complementation of the *mpl1* mutant due to lack of available genetic transformation system in the mutant background,” The fact that you have multiple independent alleles of *mpl1* that all have mutations in this gene is sufficient evidence that you have identified the correct gene.

Reviewer #2 (Remarks to the Author):

The paper entitled “Molecular Mechanisms of Acropetal Leaflet Initiation in Chickpea Pinnate Compound Leaf Patterning” offers an intriguing investigation into the development of compound leaves and the molecular mechanisms behind leaflet formation. The study focuses on the *mpl1* mutants in chickpea, which display higher-ordered pinnate leaves with a remarkably increased number of leaflets. The authors successfully identify MPL1 as a C₂H₂-zinc finger protein that plays a pivotal role in establishing a morphogenetic gradient along the proximodistal axis of early leaf primordia. This gradient is crucial for the acropetal formation of leaflets. The findings highlight the significance of MPL1 in sculpting the developmental units that contribute to the specific patterns observed in compound leaves. Furthermore, the study explores the involvement of CaLEAFY, a known regulator of leaflet initiation, and suggests that MPL1 influences the spatiotemporal expression pattern of CaLEAFY. This insight provides a deeper understanding of the regulatory network governing leaflet development. Additionally, the authors propose that the auxin signaling pathway may be modulated by MPL1, further implicating this pathway in the acropetal leaflet formation process. The experimental approach employed in this research appears comprehensive and well-designed, utilizing naturally occurring mutants to elucidate the underlying molecular mechanisms. The authors effectively present their data, including gene expression patterns and morphological analyses, supporting their conclusions. The significance of this work lies in its contribution to the broader understanding of leaf development and patterning in plants. By uncovering the role of MPL1 and its interactions with CaLEAFY and auxin signaling, the study provides novel molecular insights into the sequential progression of leaflet formation. In conclusion, the paper successfully unravels the molecular mechanisms underlying acropetal leaflet initiation in chickpea pinnate compound leaf patterning. The study's findings advance our understanding of plant development and provide a foundation for future research in leaf morphogenesis. Overall, this work represents a valuable contribution to the field and merits publication in *Nature Communications*. Although I do not find any major problems in the paper, I believe that the following points should be addressed.

(1) The authors could discuss the potential implications of their findings in the context of other plant species with compound leaves. For example, I would like to know if there are any findings of LFY overexpression phenotype, including other species. I also would like to know if there is any knowledge on the function of MPL1 orthologs in other plants.

(2) A complementation test is a good way to determine the causal gene. However, it is understandable that transformation is difficult in this system, and the results of linkage analysis and multiple allele analysis are sufficient to confirm this. It is mentioned in the discussion section that the complementation test is difficult, but in this case, the authors should honestly state that in the results section and insist that the causal gene is definitely MPL1 in light of the other results.

(3) Figure 3B should be revised. It is too small and hard to understand.

(4) The inference that CaARF23 is a key regulator based solely on the high expression of CaARF23 in *mpl1* is a bit of an overstatement. Since the public agrees that auxin has an important role in leaf morphogenesis, it is premature to conclude that MPL1 is a comprehensive regulator of morphogenesis through auxin without a detailed analysis of the relationship between MPL1 and auxin.

(5) I think the relationship between LEY and ARF should be explained in detail if there is any past knowledge of this relationship.

(6) page 19, line 10: Fig.7f  Fig. 7h

Reviewer #3 (Remarks to the Author):

In the present study, the authors identified a responsible gene for 60-years-old natural mutant of chickpea, *mpl1*, that has higher-order leaflet formation in proximal part of a leaf. They revealed that MPL1 encodes a C2Hc Zinc-Finger TF, an ortholog of PALM1 of *Medicago truncatula*. And MPL1 expression was claimed to be complementary to the LFY expression in spatial manner in the leaf primordia. The authors also showed that MPL1 binds to promoter of LFY that might suppress its expression directly.

Identification of 60-years-old classical mutant from chickpea is a good job considering that chickpea is not yet established for molecular genetic studies. The story of this manuscript highly depends on past knowledge on PALM1 and LFY homologs in the other model species. In this sense, impact of the discoveries in this study was modest.

The most unsatisfied point in this manuscript is a weakness of the supportive data for the authors' statement on spatial gradient pattern of the PALM1 expression level along the proximo-distal axis of the leaf primordia. Yes, I could recognize the spatial gradient of LFY expression level long the proximo-distal axis of the leaf primordia as the authors claimed (Figs. 5 and 6). But in my eyes, data for PALM1 in Fig. 4 is hard to support the authors' claim (Fig. 4n). Indeed, according to the quantitative analysis shown in Fig. 4b, MPL1 expression increased from P3 to P5, but did not increase its level from P5 to P7, then again increase in P8 (moreover, no statistic analysis was done here). If the MPL1 really increase its expression in correlated with addition of new leaflets to each leaf primordia as modeled in Fig. 4n, the data in Fig. 4b should not be like this. Considering that the authors selected the most 'typical' in situ figures here, this unclearness on the spatial expression cline is a frustrated point. Without being persuaded on this spatial expression cline, I cannot agree with authors on the model shown in Figure 6r.

In related to the above, I found that the gradient of expression level of MPL1 along the longitudinal axis of leaf primordia is much, much weaker than the morphological phenotype of the *mpl1* mutant in terms of higher-order leaflet formation. As seen in Figure 2 and 3, only basal 2 or 3 leaflets are converted into compound morphology in the *mpl1* mutant, suggesting some sharp border between the basal two to three 'nodes' and more upper part of leaf primordia. Why so sharp, abrupt change is observed, while gradient of the MPL1 mRNA level is much subtle?

Related to it, as authors wrote in the second page of "MPL1 negatively regulates CaLFY expression", LFY expression level became "uniformly along the longitudinal axis (Fig. 6j-l)" by the *mpl1-1* mutant. Thus, if the level of LFY is a key to control the complexity of leaflets, all the lateral leaflets should convert into complex morphology in an equal level in *mpl1*. But

only a few basal leaflets changed the morphology in *mpl1-1*. Why?

In Discussion the authors might have tried to explain the inconsistency between the smooth gradient in the MPL1 expression level and abrupt, sharper gradient in the leaflet complexity in wild type by introducing an idea of 'threshold'. But this is just an interpretation. Moreover, I found a queer point in the Fig. 3e. While *mpl1-2* mutant shows smooth cline in the complexity of leaflet structure along the proximo-distal axis, *mpl1-3* does not. In the *mpl1-3*, the most basal is most complex, but second basal is simple; and the third basal is secondarily complex showing intermediate shape between the most basal and the second basal. Because *mpl1-3* is thought to be equally null allele as *mpl1-2*, this non-straight, non-smooth cline along the longitudinal axis in the *mpl1-3* seems to be hard to be explained by a simple spatial gradient pattern of the PALM1 expression level.

And in the section "MPL1 is the orthologous gene of PALM1", I felt that the mutation phenotype of *M. truncatula palm1* is quite weaker than the *mpl1* of chickpea (Supplementary Fig. 9). Why, the authors think, does *mpl1* show much stronger and severer phenotype: higher order leaflet formation in the *mpl1* mutant, than in *palm1*?

Minor comments

1. Third page, center of Introduction: "a vey" reads " a very".
2. Results, fifth line of "MPL1 negatively regulates CaLFY expression": 'apexes' reads "apices".
3. The first sentence of the "MPL1 negatively regulates CaLFY expression" section: the authors are requested to add reference(s) for their statement here.
4. Role of LFY. In this manuscript, all the story depends on an idea that LFY expression equals to compound leaf formation. But the authors wrote that LFY is also detected in the stipule primordia (Fig. 5e,f), while stipule does not show 'compound phenotype'. This fact indicates that expression of LFY does NOT always result in complex morphogenesis; the authors' story that heavily depend on the data of expression level is not solid.

REVIEWER COMMENTS

Reviewer #1 (Remarks to the Author):

The development of compound leaves has long fascinated developmental biologists because it informs our fundamental understanding of the trade off between differentiation and the maintenance of undifferentiated cells with the potential to generate new structures. In the manuscript entitled "Control of compound leaf patterning by MULTI-PINNATE LEAF1 (MPL1) in chickpea" Liu et al. beautifully reveal how a gradient between CaLFY maintaining undifferentiated status from the tip downward and MPL1 promoting differentiation from the base upward controls the initiation of leaflets. They first show stunning SEM images which reveal the acropetal (base to tip) development of leaflets in the wild type chickpea leaf and how this pattern is enhanced in *mpl1* mutants where leaflet form on the lower leaflets. The acropetal development of leaflets is interesting because it is the reverse of other well studied compound leaf systems such as tomato and Cardamine in which leaflets develop basipetally. The authors positionally clone the MPL1 gene and find it encodes a C2H2 zinc finger transcription factor ortholog of PALM1, which is known to regulate leaflet number in *Medicago*. With beautiful in situ hybridization images, the authors show that MPL1 is expressed in developing leaflets starting from the base and moving upward as the leaflets mature. In basipetal leaflet systems, the expression of KNOX genes (STM and BP) promotes the maintenance of the undifferentiated state. Similar to pea, the authors show that CaSTM and CaBP1 are expressed only in the meristem, so they are not candidate factors. Again similar to Pea they find that CaLFY is expressed in developing leaves, particularly in the undifferentiated tip and youngest leaflets. CaLFY is strongly upregulated in *mpl1* mutants, and the authors show that MPL1 can bind to the CaLFY promoter using EMSAs, suggesting that MPL1 directly represses CaLFY expression, this promoting differentiation. Finally the authors examine auxin transport because auxin transport has been shown to promote leaflet and serration formation in other studies. The author's treatment of wild type with auxin transport inhibitor NPA shows that auxin transport does promote leaflet and serration formation. However, the interaction with *mpl1* is complex because NPA increases leaflet formation in the mutant. CaARF23 which is normally downregulated in NPA treated wild type plants is not downregulated in NPA treated *mpl1* mutants. I suggest the authors use caution in interpreting the NPA results. Overall, the manuscript is well written, the figures are beautiful, and the experiments are well done with appropriate controls. I think this manuscript adds an important piece to our understanding of compound leaf development and the different ways plants balance differentiation and maintaining undifferentiated tissues.

Response: Thank the reviewer very much for the great summary of our work and for highlighting positive aspects in our work, which is very helpful in making this manuscript

better.

Minor comments:

“(ii) auxin polar transport is connected to the mechanism of MPL1 function in controlling the spatio-temporal expression of *CaARF23* and *CaLFY* during leaf development.” The data do not support this interpretation. I believe here the authors should use caution and just point to the complex interaction. It will take another whole paper worth of experiments to understand this interaction better, which is for the future.

Response: Thank you for the helpful and valuable comments. We appreciate the reviewer's insights and acknowledge the limitations of our study in elucidating the complex relationship between *MPL1*, *CaLFY*, and auxin signaling pathways during compound leaf development. We have addressed your concerns and provided the necessary revisions.

Previous studies have reported that NPA-treatment significantly inhibits leaflet production in various plant species, such as tomato (Koenig et al., 2009, Development), *C. hirsuta* (Barkoulas et al., 2008, Nature Genetics), *M. truncatula* (Zhou et al., 2011, The Plant Cell), and Arabidopsis *jk* mutant with super-compound leaves (Challa et al., 2021, Nature Plants). Consistent with these findings, we found that NPA-treatment inhibited leaflet formation in WT chickpea plants and was accompanied by a significant downregulation of *CaARF23* and *CaLFY* expression (Fig. 7f,g in the main text).

However, NPA-treatment showed a potent increase in leaflet production in the *mpl1-1* mutant, which may be attributed to several reasons. Firstly, the extremely high expression of *CaARF23* and *CaLFY* in *mpl1-1* (Fig. 7f,g in the main text) remained unaffected by the treatment. Additionally, auxin transport has been shown to play a crucial role in promoting differentiation processes (Wulf et al., 2018, Annals of Botany). Perturbing auxin transport may enhance the undifferentiated potential of leaf and leaflet primordia, resulting in increased leaflet numbers. Furthermore, a potential mechanistic link between LFY and auxin signaling pathways has been demonstrated during floral development in Arabidopsis (Yamaguchi et al., 2013, Developmental Cell). Therefore, direct mechanistic links may exist between the *MPL1-CaLFY* module and the auxin signaling pathway during compound leaf development. However, unraveling the intricate interactions between the *MPL1-CaLFY* and auxin signaling pathways will require further extensive experimentation, which could be pursued in future studies.

“We found that the NPA-treatment significantly down-regulated the *CaARF23* and *CaLFY* expression in WT, but did not affect the extremely high expression levels of these genes in *mpl1-1* (Fig. 7f).” The figure referenced should be 7h.

Response: Thanks for point out it. We have corrected it.

“By contrast, the *mpl1* mutation did not change its own expression, and the NPA-treatment

slightly down-regulated the MPL1 expression in both WT and mpl1 (Fig. 7f).” Again the figure referenced should be 7h.

Response: Thanks for point out it. We have corrected it.

“Although we did not perform complementation of the mpl1 mutant due to lack of available genetic transformation system in the mutant background,” The fact that you have multiple independent alleles of mpl1 that all have mutations in this gene is sufficient evidence that you have identified the correct gene.

Response: Thank you for giving this suggestion. We have re-organized the words in this part.

Reviewer #2 (Remarks to the Author):

The paper entitled “Molecular Mechanisms of Acropetal Leaflet Initiation in Chickpea Pinnate Compound Leaf Patterning” offers an intriguing investigation into the development of compound leaves and the molecular mechanisms behind leaflet formation. The study focuses on the mpl1 mutants in chickpea, which display higher-ordered pinnate leaves with a remarkably increased number of leaflets. The authors successfully identify MPL1 as a C2H2-zinc finger protein that plays a pivotal role in establishing a morphogenetic gradient along the proximodistal axis of early leaf primordia. This gradient is crucial for the acropetal formation of leaflets. The findings highlight the significance of MPL1 in sculpting the developmental units that contribute to the specific patterns observed in compound leaves. Furthermore, the study explores the involvement of CaLEAFY, a known regulator of leaflet initiation, and suggests that MPL1 influences the spatiotemporal expression pattern of CaLEAFY. This insight provides a deeper understanding of the regulatory network governing leaflet development. Additionally, the authors propose that the auxin signaling pathway may be modulated by MPL1, further implicating this pathway in the acropetal leaflet formation process. The experimental approach employed in this research appears comprehensive and well-designed, utilizing naturally occurring mutants to elucidate the underlying molecular mechanisms. The authors effectively present their data, including gene expression patterns and morphological analyses, supporting their conclusions. The significance of this work lies in its contribution to the broader understanding of leaf development and patterning in plants. By uncovering the role of MPL1 and its interactions with CaLEAFY and auxin signaling, the study provides novel molecular insights into the sequential progression of leaflet formation. In conclusion, the paper successfully unravels the molecular mechanisms underlying acropetal leaflet initiation in chickpea pinnate compound leaf patterning. The study's findings advance our understanding of plant development and provide a foundation for future research in leaf morphogenesis. Overall, this work represents a valuable contribution to the

field and merits publication in Nature Communications. Although I do not find any major problems in the paper, I believe that the following points should be addressed.

Response: We are grateful to the reviewer for these supportive comments.

(1) The authors could discuss the potential implications of their findings in the context of other plant species with compound leaves. For example, I would like to know if there are any findings of LFY overexpression phenotype, including other species. I also would like to know if there is any knowledge on the function of MPL1 orthologs in other plants.

Response: Thank you for your valuable feedback, which has greatly improved our paper. We have incorporated the suggested revisions and expanded our discussion on LFY orthologs in leaf development across different species and the functions of MPL1 orthologs and homologs of various plants.

On the one hand, we have provided a systematic discussion on LFY orthologs and their role in leaf development across various species.

Although the maintenance of undifferentiated status in early compound leaf primordia by KNOXI genes is nearly ubiquitous across seed plants, it is now recognized that LFY genes are widely required in this process across different species (below: Fig 1; Fig 2A,B). In the fern *Ceratopteris richardii*, both *CrKNOXI* and *CrLFY* genes were expressed in the complex frond (leaf-like) primordia of the young sporophyte, and suppression of *CrLFY* activity by RNAi resulted in simple fronds (Plackett et al, 2018, elife). This suggested an ancestral function of LFY in maintaining the undifferentiated tissues of vegetative organs.

In most non IRLC legumes (outside of the IRLC), *KNOXI* genes were expressed in their compound leaves (Champagne et al, 2007, Plant Cell), but *LFY* orthologs appear to also have important roles (above: Fig 1). In mung bean, the loss-of-function mutant of the *LFY* gene *UNIFOLIATE LEAF (UN)* resulted a completely simple leaf phenotype (Jiao et al, 2019, Horticulture Research; below: Fig 2C). In *L. japonicus* (Wang et al, 2013, Journal of Integrative Plant Biology; below: Fig 2D) and soybean (Champagne et al, 2007, Plant Cell), the loss-of-function mutation or RNAi silencing resulted in reduced leaflet number.

Fig 2
In IRLC legumes, the expression of *KNOXI* genes is absent in leaf primordia, whereas the *LFY* orthologs, such as *UNI* in pea and *SGL1* in *M. truncatula*, are expressed in leaf primordia (above: Fig 2E,F). The loss-of-function mutants *sgl1* and *uni* completely develop simple-like leaves (Hofer et al, 1997, Current biology; Wang et al, 2008, Plant Physiology), while those mutants of the two species with increased leaflet production showed great upregulation in *LFY* genes (Gourlay et al, 2000, Plant Cell; He et al, 2020, nature plants) (above: Fig 2E,F). Therefore, *LFY* is considered to completely take place of the role of *KNOXI* genes in compound leaf development in IRLC legumes. Our study strengthens this idea and meanwhile suggests that *LFY* activity is subjected to delicate spatiotemporal regulation thus to form a specific compound leaf pattern.

Recent works in *M. truncatula* revealed that *KNOXI* can complement the phenotype of *sgl1* mutant (Pautot et al, 2022, IJMS), and their overexpression dramatically increases leaflet production (Zhou et al, 2014, Plant Cell). However, overexpression of *SGL1* (*35S::SGL1*) alone has no effect on the compound leaf pattern (Zhou et al, 2014, Plant Cell); this may be attributed to that a finely differential expression pattern of *SGL1* in an early compound leaf primordium is necessary for the normal program of balance differentiation and maintaining undifferentiated tissues, and there are lots of regulators to fine-tune the *SGL1* activity. Moreover, in *C. hirsuta*, loss or gain of *LFY* function only affects the progression of leaf heteroblasty (Monniaux et al, 2017, New phytologist). These indicate that *LFY* acts to specify an undifferentiated state in a context-dependent manner, requiring interaction with other factors, different from *KNOXI* which maintains an undifferentiated state despite developmental context.

On the other hand, we also comprehensively discussed the functions of *MPL1* orthologs in regulating organ morphogenesis in other plants as follows.

The *M. truncatula* ortholog of *MPL1*, *PALM1*, specially promotes the differentiation of early LL primordia by directly repressing *SGL1* expression (Chen et al, 2011, PNAS). The *Aquilegia coerulea* ortholog *POP* is critical for the development of floral nectar spurs and also contributes to

compound leaf development (Ballerini et al, 2020, PNAS). Knock-down of POP by VIGS could result in exaggerated dissections between leaflet lobes such that the overall leaf architecture shifted from ternately compound to biternately compound leaves. Phylogenetic analysis showed MPL1, PALM1 and POP1 formed a distinct clade sister to the RBE and SUP clades (Supplementary Fig. 8). In the RBE clade, *A. thaliana* *RABBIT EARS* (*RBE*) promotes cell proliferation at the boundaries of petal primordia by controlling the cell fate transition from mitotic growth to differentiation (Takeda et al, 2004, Development), while the pea *STIPULES REDUCED* (*St*) regulates both cell division and cell expansion involved in stipule development (Moreau et al, 2018, New phytologist). The *rbe* mutant have reduced numbers of petals and conversion of petals to filaments; the *st* mutant unusually converts the large stipules of pea into smaller organs. *A. thaliana* *SUP* and rice *SMALL REPRODUCTIVE ORGANS* (*SRO*) formed another clade sister to *MPL1/PALM1/POP* and *RES/St*, both of which affect cell division in flower development. In *sup* mutants, derepressed local YUC1/4 activity elevates auxin levels at the boundary between whorls 3 and 4, which leads to an increase in the number and the prolonged maintenance of floral stem cells, and consequently an increase in the number of reproductive organs (Moreau et al, 2018, Embo Journal). The *sro* mutant developed defective flowers with reduced size in stamens and pistils and no viable pollen grains (Xu et al, 2022, New phytologist). These findings suggested that MPL1 homologs have similar effects in organ morphogenesis, specifically in regulating cell division.

(2) A complementation test is a good way to determine the causal gene. However, it is understandable that transformation is difficult in this system, and the results of linkage analysis and multiple allele analysis are sufficient to confirm this. It is mentioned in the discussion section that the complementation test is difficult, but in this case, the authors should honestly state that in the results section and insist that the causal gene is definitely MPL1 in light of the other results.

Response: We have revised the section according to your helpful suggestions, rewriting that “*Ca_02268* is definitely the *MPL1* gene.”

(3) Figure 3B should be revised. It is too small and hard to understand.

Response: We have revised the section according to your helpful suggestions. Figure 3B represents the enlargement of the peak of the BSA mapping interval. Each triangle represents an annotated gene, with blue indicating genes carrying SNP mutations and red indicating the most likely candidate gene *Ca_02268* that simultaneously carried SNP and deletion mutation.

(4) The inference that CaARF23 is a key regulator based solely on the high expression of CaARF23 in *mpl1* is a bit of an overstatement. Since the public agrees that auxin has an important role in leaf morphogenesis, it is premature to conclude that MPL1 is a comprehensive regulator of morphogenesis through auxin without a detailed analysis of the

relationship between MPL1 and auxin.

Response: Thank the reviewer for bringing up this point. We agree with the opinions of both reviewer 1 and reviewer 2, and recognize the need to revise our interpretation regarding the complex interaction between MPL1 and the auxin pathway. We have revised the relevant section based on your helpful suggestions. The revised statement now highlights that “the MPL1-CaLFY module is integrated with the auxin signaling pathway to regulate compound leaf morphogenesis.”

(5) I think the relationship between LFY and ARF should be explained in detail if there is any past knowledge of this relationship.

Response: We sincerely thank the referee for providing this valuable suggestion. We have carefully revised our manuscript to provide a more detailed explanation of the relationship between LFY and ARF, as well as past knowledge related to this relationship.

In *Medicago slm1* mutants, which contain loss-of-function mutations in the ortholog of Arabidopsis *PIN1*, a reduction in the number of lateral leaflets is associated with a downregulation of *SGL1* expression (Zhou et al, 2011, Plant Cell), suggesting a relationship between the LFY and auxin pathways. In Arabidopsis flower development, the ARF MONOPTEROS (MP/ARF5) directly induces expression of *LFY* via evolutionarily conserved and biologically important cis-regulatory motifs in the *LFY* promoter; *LFY* also positively feeds back to the auxin pathway (Yamaguchi et al, 2013, Developmental Cell). However, to the best of our knowledge, there is no other published information about the relationship between LFY and ARF in compound leaf development.

In our study, multiple ARF genes, including the MP ortholog *CaARF11*, are upregulated in *mpl1*. The NPA-treatment inhibited the leaflet formation in chickpea WT plants, accompanied by a significant downregulation of *CaARF23* and *CaLFY* (Fig. 7f,g). However, the treatment potently increased the leaflet production in *mpl1-1*. This is may associated with multiple reasons. At first, the extremely-high expression of *CaARF23* and *CaLFY* (Fig. 7f,g) was not unaffected by the NPA-treatment. Secondly, auxin transport play an important role of in promoting the differentiation process (Wulf et al, 2019, Annals of Botany). Perturbing auxin transport may enhance the undifferentiated potential of leaf and leaflet primordia, resulting in increased leaflet numbers. Furthermore, potential mechanistic links between LFY, ARF and auxin transport may exist during compound leaf development. We have addressed these in the corresponding section.

However, unraveling the intricate interactions between the MPL-CaLFY module, ARF regulators and auxin signaling pathways will require further extensive experimentation, which could be pursued in future studies.

(6) page 19, line 10: Fig.7f  Fig. 7h

Response: Thanks for point out it. We have corrected it.

Reviewer #3 (Remarks to the Author):

In the present study, the authors identified a responsible gene for 60-years-old of chickpea, *mpl1*, that has higher-order leaflet formation in proximal part of a leaf. They revealed that *MPL1* encodes a C2Hc Zinc-Finger TF, an ortholog of *PALM1* of *Medicago truncatula*. And *MPL1* expression was claimed to be complementary to the *LFY* expression in spatial manner in the leaf primordia. The authors also showed that *MPL1* binds to promoter of *LFY* that might suppress its expression directly.

Identification of 60-years-old classical mutant from chickpea is a good job considering that chickpea is not yet established for molecular genetic studies. The story of this manuscript highly depends on past knowledge on *PALM1* and *LFY* homologs in the other model species. In this sense, impact of the discoveries in this study was modest.

Response: We are glad that Reviewer 3 appreciated the manuscript as a good molecular genetic study of the non-model legume species Chickpea. Nowadays, the identification of completely new genes from non-model plants has become relatively rare. Instead, there is a growing trend that genes cloning from some classical genetic mutants are often turned out to be homologous to known genes. However, there is a world-wide interest in understanding how these conserved genes exhibit specific functional patterns in different biological processes and across various species.

Although the *MPL1* gene is orthologous to the known *PALM1* of *Medicago*, our study revealed a functional manner of *MPL1* distinct from *PALM1* in control leaflet patterning. This finding adds an important piece to our understanding of the different ways that plants balance differentiation and maintaining undifferentiated tissues and thus form different patterns during compound leaf development.

The most unsatisfied point in this manuscript is a weakness of the supportive data for the authors' statement on spatial gradient pattern of the *PALM1* expression level along the proximo-distal axis of the leaf primordia. Yes, I could recognize the spatial gradient of *LFY* expression level along the proximo-distal axis of the leaf primordia as the authors claimed (Figs. 5 and 6). But in my eyes, data for *PALM1* in Fig. 4 is hard to support the authors' claim (Fig. 4n). Indeed, according to the quantitative analysis shown in Fig. 4b, *MPL1* expression increased from P3 to P5, but did not increase its level from P5 to P7, then again increase in P8 (moreover, no statistic analysis was done here). If the *MPL1* really increase its expression in correlated with addition of new leaflets to each leaf primordia as modeled in Fig. 4n, the data in Fig. 4b should not be like this. Considering that the authors selected the most 'typical' in situ figures here, this unclearness on the spatial expression cline is a frustrated point. Without being persuaded on this spatial expression cline, I cannot agree with authors on the model shown in Figure 6r.

Response: Thank the reviewer for bringing these concerns to our manuscript. We have carefully studied the comments, and made major modifications to the manuscript including restructuring and rewriting passages, with particular attention given to the discussion section.

Based on our results and the reviewer's concerns, we summarized the spatial expression pattern of *PALM1* along the proximo-distal axis of the leaf primordia into three major aspects:

(1) *MPL1* expression was significantly higher in the proximal portion (PP) than the distal portion (DP) in the relatively young primordia (P4~P5) (Figure 4c in main text). However, in the more differentiated primordia (P6~P8), although the *MPL1* expression was increased compared to younger primordia, but there was no significant difference between the PP and DP (Figure 4c in main text).

(2) In situ results showed that, during the process of LL initiation (P2~P4 stages), *MPL1* expression is seen throughout each differentiated LL primordium, but barely detected in the undifferentiated tip; more importantly, during this process, *MPL1* expression shows a spatio-temporal pattern that starts from the most basal LL primordia and progresses acropetally as leaflet primordia sequentially formed.

(3) Most importantly, we acknowledge that, once *MPL1* is expressed, the intensity of *MPL1* in situ signals may do not differ between an equal number of cells of two LL primordia. However, because the LL primordia show an acropetally decreasing gradient in size, it is reasonable that the more differentiated LL primordia consistently have an earlier and broader expression of *MPL1* than younger ones. Accordingly, we have made revisions to the Figure 4n and 6r, incorporating purple diagonal stripes to indicate the expression of *MPL1*. This provides a visual representation of the spatial distribution of *MPL1* expression in the compound leaf primordia.

By the way, we chosen the in situ hybridization images based on several criteria. Firstly, the chosen image should feature a well-structured compound leaf primordium, making it easier to identify the organ pattern and to determine its developmental stage. Secondly, the in situ signals should be fully proved by serial sections, which were comprehensively presented in Supplementary Figures.

Then, we don't think our qRT-PCR data in Fig. 4b are in conflict with the modeled expression pattern in Fig. 4n. The model Fig. 4n is mainly based on the in situ hybridization results and specially depicted a spatio-temporal pattern of *MPL1* expression in the compound leaf primordia from stages P1 to P5. In this model, *MPL1* expression starts from the most basal LL primordia during an early P3 stage, progresses acropetally as leaflet primordia sequentially formed during stages from P3 to P4, and lastly extended to the TL primordium during the P5 stage. Then, we performed a replicate qRT-PCR experiment and conducted statistical analysis. The results showed that the expression trend of *MPL1* during leaf development is largely consistent with the original data that *MPL1* expression increased from P3 to P5, consistent with its modeled pattern in Fig.4n. However, the expression pattern of *MPL1* during later stage after P6 were not included in the model, because that is similar to the P5~P6 stage pattern. This is evident by the fact that the *MPL1* expression did not increase its level continuously with time from P5 to P8.

In related to the above, I found that the gradient of expression level of *MPL1* along the longitudinal axis of leaf primordia is much, much weaker than the morphological phenotype of the *mpl1* mutant in terms of higher-order leaflet formation. As seen in Figure 2 and 3, only basal 2 or 3 leaflets are converted into compound morphology in the *mpl1* mutant, suggesting some sharp border between the basal two to three 'nodes' and more upper part of leaf primordia. Why so sharp, abrupt change is observed, while gradient of the *MPL1* mRNA level is much subtle?

Response: We are very sorry for any confusion caused. We agree with the review that, in the original version of our modeled Figure 4n and 6r, the different darkness levels of purple cannot really reflect the expression of *MPL1*. However, as mentioned above, *MPL1* expression shows a spatio-temporal pattern that starts from the most basal LL primordia and progresses acropetally as leaflet primordia sequentially formed. Because the size of the LL primordia exhibited a acropetally decreasing gradient, it is reasonable that the more differentiated LL primordia consistently have an earlier and broader expression of *MPL1* than younger ones. We thus have made revisions to the Figure 4n and 6r, incorporating purple diagonal stripes to indicate the expression of *MPL1*. We think that the revised Figure provides a visual representation of the spatial distribution of *MPL1* expression in the compound leaf primordia.

The consistent correlation between the expression pattern of *MPL1* and the morphological phenotype after its mutation can be understood as follows.

The maintenance of the acropetal leaflet initiation in chickpea involves a morphogenetic gradient along the proximodistal axis of early leaf primordia, with young undifferentiated cells at the tip and more differentiated cells at the base. We suggested that this gradient is related to the molecular interaction between *MPL1* and *CaLFY*. *CaLFY* is highly expressed in the tip of the whole compound leaf primordium, acting to maintain an undifferentiated status, but its expression is significantly lower in the basal differentiated LL primordia. *MPL1* is barely expressed in the undifferentiated tip, but shows consistent expression throughout each differentiated LL primordium, where it repress the *CaLFY* expression. The below Fig 3a-d illustrates the process of leaflet initiation (P2~P4) in WT. The tips of compound leaf primordia maintain an undifferentiated status (green); the *MPL1* expression promotes the differentiation of LL primordia (purple diagonal stripes).

In the *mpl1* mutant, the loss-of-function of *PALM1* resulted in that the LL primordia have a high activity of *CaLFY* and thus were converted as "pseudo undifferentiated tips" (below: Fig 3i). We considered this conversion follows a developmental gradient manner (below: Fig 3e-g). In detail, as shown in the above figure, the most basal LL primordia were converted as an early "pseudo undifferentiated tip" during an early stage upon their initiation (below: Fig 3f), which has stronger capability to generate new leaflet primordia, whereas the more distal LL primordia were converted as a late "pseudo undifferentiated tip" (below: Fig 3g), which soon later was differentiated into a leaflet primordium and thus has less potential to generate new primordia; consequently, the more

proximal first-order leaflets can produce more higher-ordered leaflets.

Related to it, as authors wrote in the second page of “MPL1 negatively regulates CaLFY expression”, LFY expression level became “uniformly along the longitudinal axis (Fig. 6j-l)” by the *mpl1-1* mutant. Thus, if the level of LFY is a key to control the complexity of leaflets, all the lateral leaflets should convert into complex morphology in an equal level in *mpl1-1*. But only a few basal leaflets changed the morphology in *mpl1-1*. Why?

Response: As mentioned above, in the *mpl1* mutant, the absence of *MPL1* converted the LL primordia as “pseudo undifferentiated tips”, and this conversion follows a developmental gradient manner. In detail, the most basal LL primordia were converted as the early “pseudo undifferentiated tip” (above: Fig 3f) upon initiation, which has stronger capability to generate new leaflet primordia, whereas the more distal LL primordia were converted as the late “pseudo undifferentiated tip” (above: Fig 3g), which soon later was differentiated into the TL primordium and thus has less potential to generate new primordia; consequently, the more proximal first-order leaflets can produce more higher-ordered leaflets.

Moreover, it cannot be disregarded other regulators to maintain a robust differentiation program for leaflet primordia. In WT, although *MPL1* is barely expressed in the tip over time during early stages (P2~P4), the undifferentiated status of the tip still gradually decreases, indicating other regulator promoting differentiation.

In Discussion the authors might have tried to explain the inconsistency between the smooth gradient in the *MPL1* expression level and abrupt, sharper gradient in the leaflet complexity in wild type by introducing an idea of ‘threshold’. But this is just an interpretation. Moreover,

I found a queer point in the Fig. 3e. While *mpl1-2* mutant shows smooth cline in the complexity of leaflet structure along the proximo-distal axis, *mpl1-3* does not. In the *mpl1-3*, the most basal is most complex, but second basal is simple; and the third basal is secondarily complex showing intermediate shape between the most basal and the second basal. Because *mpl1-3* is thought to be equally null allele as *mpl1-2*, this non-straight, non-smooth cline along the longitudinal axis in the *mpl1-3* seems to be hard to be explained by a simple spatial gradient pattern of the PALM1 expression level.

Response: Thank you for your valuable feedback regarding Figure 3e in our original manuscript. We appreciate the reviewers' comments and would like to address the confusion surrounding the phenotypes of *mpl1-2* and *mpl1-3* mutants. The seeds for both alleles were obtained by the co-author Pro. Million from the USDA and were brought to China on April 18th, 2023. On May 20th, compound leaves of about 5-week-old plants were photographed. At that time, the plants had produced no more than 11 mature leaves. In our original manuscript, we specifically chose to capture images of fully mature compound leaves at the L8 node, while the leaves above that node were still in the process of expanding.

As shown in below Fig 4, the mutant exhibits a strong heteroblastic change in leaflet number during its early vegetative growth. As a result, compound leaves at the L8 node varied significantly among different alleles and individuals. To address this issue, we have made revisions in the manuscript. Specifically, we replaced Figure 3e with images of mature compound leaves at the L12 node, which provides a more representative sampling of the leaflet structures. We apologize for any confusion caused by our initial description and think that the revised manuscript will provide a clearer understanding of the observed leaflet structures.

And in the section “MPL1 is the orthologous gene of PALM1”, I felt that the mutation phenotype of *M. truncatula palm1* is quite weaker than the *mpl1* of chickpea (Supplementary Fig. 9). Why, the authors think, does *mpl1* show much stronger and severer phenotype: higher order leaflet formation in the *mpl1* mutant, than in *palm1*?

Response: Chickpea leaf development has a long period of leaflet initiation, accompanied by a sustained, high, and broad expression of *CaLFY*. However, trifoliolate leaf development in *M. truncatula* involves only once event of leaflet initiation (below Fig 5,6); transient and restricted strong expression of *SGL1* maintained a short period of undifferentiated status in the tip of the early compound leaf primordium (below Fig 5a-d; Fig 6a-c). In the *palm1* mutant, the earliest two LL primordia acquire a temporary elevated expression of *SGL* (below Fig 5e, blue arrows) and thus were converted as “pseudo undifferentiated tips” (below Fig 6f); they therefore possess a limited potential to initiate one new leaflet primordium during the later stage (P3) (below Fig 6g).

Fig 5

Fig 6

In fact, in this mutant, when all five leaflet primordia are formed (P4), each LL primordium was further comparable to “a pseudo differentiated TL primordium”, with an expression pattern of *SGL1* similar to that of the TL primordium (above: Fig 5f,6g). This is evident when the *palm1* mutant was combined with the *pinna1* mutant that disrupts the gene encoding a BEL1-like homeodomain protein. In the *palm1 pinna1* double mutant, the first-order LLs were converted as “pseudo TLs” by

the *palm1* single mutation and thus additional “second-order LLs” were formed due to the *pinna1* mutation (below: Fig 7d).

Minor comments

1. Third page, center of Introduction: “a vey” reads “ a very”.

Response: Thank you for your careful reading, and we have made the necessary correction.

2. Results, fifth line of “MPL1 negatively regulates CaLFY expression”: ‘apexes’ reads “‘apices’.

Response: Thank you for your careful reading, and we have made the necessary correction.

3. The first sentence of the “MPL1 negatively regulates CaLFY expression” section: the authors are requested to add reference(s) for their statement here.

Response: Thank you for your careful reading, and we have made the necessary correction.

4. Role of LFY. In this manuscript, all the story depends on an idea that LFY expression equals to compound leaf formation. But the authors wrote that LFY is also detected in the stipule primordia (Fig. 5e,f), while stipule does not show ‘compound phenotype’. This fact indicates that expression of LFY does NOT always result in complex morphogenesis; the authors’ story that heavily depend on the data of expression level is not solid.

Response: Thanks for bringing up these important points. The view that the LFY orthologs function in place of KNOXI genes in the development of compound leaves in IRLC legumes is supported by a lot of previous studies (Hofer et al, 1997, Current biology; Champagne et al, 2007, The Plant Cell; Wang et al, 2008, Plant Physiol; Zhou et al, 2014, The Plant Cell). The *KNOXI* gene is found to be nearly ubiquitous across seed plants genes to maintain the undifferentiated status of leaf primordia of compound-leaved species (above: Fig 1). In IRLC legumes, the *KNOXI* genes are not expressed in leaf primordia, and instead, the *LFY* orthologs, known as *UNI* in pea and *SGL1* in *M. truncatula*, are expressed. Loss-of-function mutants of the *FLO/LFY* orthologs *SGL1* in *M. truncatula* and *UNI* in pea completely develop simple-like leaves (above: Fig 1, Fig 2). In *Medicago*, *KNOXI* can complement the *sgl1* mutant and *KNOXI*-overexpression led to increased leaflet production (Pautot et al, 2022, IJMS). Chickpea belongs to IRLC legumes. Our study showed that the *CaLFY* rather than *KNOXI* was associated with the undifferentiated status of the leaf primordia, and thus strengthened the view.

We really appreciate the review concerning the stipule development, which has attracted a high level of interest. We are currently conducting related studies in this topic and are exploring the roles of BOP and ST (homologous to MPL1, see supplementary Fig 8) in this process. In present study, the observations the LFY is also detected in the stipule primordia while stipule does not show 'compound phenotype' can be explained by multiple reasons as follows.

At first, compared to WT, *mpl1-1* showed stronger expression of *CaLFY* in leaflet primordia, particularly in those proximal leaflet primordia (above: Fig 8 and 9). However, in stipule primordia, there was no obvious difference in the pattern and intensity of *CaLFY* expression between WT and *mpl1*; both genotypes showed weak *CaLFY* expression in the stipule primordia at different developmental stages (above: Fig 8 and 9).

Secondly, the stipule follows a developmental program significantly different from that of leaflets. Stipule primordia initiate opposite to the proximal end of the compound leaf primordium, and undergo faster differentiation than the remaining portion of the compound leaf primordium. Studies in pea and *M. truncatula* revealed that the orthologs of BLADE-ON-PETIOLE (BOP) play a conserved and critical role in the control of stipule development (Couzigou et al, 2012, Plant Cell; Zhang et al, 2022, New Phytologist). The loss-of-function mutant *coch* of pea, the, led to a conversion of a normal stipule into a compound leaf-like structure. COCH1 could recognize the *UNI* promoter in protoplasts and repress its expression. *UNI* mRNA was not detected in stipule primordia in pea WT, but was observed in that of the *coch1* mutant. In *M. truncatula*, MtNROOT1 and MtNROOT2 are orthologs of BOP. The *mntoot1 mntoot2* double mutant also converts stipules into compound leaf-like structures (Zhang et al, 2022, New Phytologist). MtNROOTs directly bind to the *SGL1* promoter and inhibit its transcription. *SGL1* expression was clearly observed in the developing leaf-like stipule primordia of the *mntoot1 mntoot2* double mutant, but not detected in the normal stipule primordia of WT. Both COCH1, MtNROOT1 and MtNROOT2 are specially expressed in the stipule primordia, but not in the other portion of the compound leaf primordium (Zhang et al, 2022, New Phytologist).

Therefore, the specification and maintenance of the identity of stipule primordia involves specific mechanisms different from that responsible for the leaflet formation. Nevertheless, LFY orthologs still have potential roles in the development of stipules, because upregulation of LFY expression is one reason leading to the transition from stipules to compound leaf-like structure in *coch* and *mntoot1 mntoot2* mutants.

Thirdly, LFY expression to maintain an undifferentiated status is a necessary but not sufficient condition for the leaflet formation in IRLC legumes. In *M. truncatula*, overexpression of *SGL1* (*35S::SGL1*) has no effect on the compound leaf pattern (Zhou et al, 2014, Plant Cell), indicating that the role of *SGL1* in promoting leaflet initiation is developmental context-dependent. In other words, the formation of leaflet primordia involves other necessary regulatory mechanisms, such as NAM/CUC-mediated boundary formation between leaflets and polar auxin transport creating auxin maxima (Blein et al., 2008, Science) (below: Fig 10A). The silencing or mutation of *CUC/NAM* in different species all resulted in different degrees of fusion and decrease in the number of leaflets, whereas ectopic expression of *CUC/NAM* genes resulted in a compound leaf phenotype of increased leaflet number (Blein et al., 2008; Berger et al., 2009; Wang et al., 2013; Rast-Somssich et al., 2015; Jiao K. Y. et al., 2019) (below: Fig 10B-G). Auxin is essential for the initiation of leaf common primordia from the flanks of SAM and leaflet primordia from the margins of early compound leaf primordia. Loss-of-function mutations in *PIN1* orthologs, an important auxin efflux

transporter, resulted in severe defects in leaf development, such as reduced marginal serrations, fusion leaflets, reduced leaf production and reduced leaflet number (below: Fig 10H,J,K).

In conclusion, the development of the proximal stipules and distal leaflets involves different mechanisms, especially completely different mechanisms to repress LFY expression. Moreover, the expression of LFY is a necessary but not sufficient condition for the developmental change from a simple primordium to a complex structure of several leaflet primordia for IRLC legumes. Many other factors are also necessary for this process.

REVIEWERS' COMMENTS

Reviewer #1 (Remarks to the Author):

The authors have thoroughly addressed all of my comments. I commend them on their well written manuscript, exciting results, and beautiful figures.

Reviewer #2 (Remarks to the Author):

The author appropriately addressed all of my comments.

Reviewer #3 (Remarks to the Author):

I respect the author's efforts made for the present revision. Nearly all the concerns were resolved now. In particular, descriptions on the spatio-temporal expression pattern were greatly improved and now are agreeable.

Only one point: The authors wrote that the MPL1/PALM1 is lost in the monocot lineage. Because this is interesting point and I examined it by BLAST search and found that some monocot genomes have this MPL1/PALM1 clade, too. I think that the choice the authors made accidentally were not representative to all the monocot clade. For example, please search for Banana. The authors will soon find that some clades of monocot lost this gene, but some monocots still keep this in their genome.

Reviewer #3 (Remarks to the Author):

I respect the author's efforts made for the present revision. Nearly all the concerns were resolved now. In particular, descriptions on the spatio-temporal expression pattern were greatly improved and now are agreeable.

Only one point: The authors wrote that the MPL1/PALM1 is lost in the monocot lineage. Because this is interesting point and I examined it by BLAST search and found that some monocot genomes have this MPL1/PALM1 clade, too. I think that the choice the authors made accidentally were not representative to all the monocot clade. For example, please search for Banana. The authors will soon find that some clades of monocot lost this gene, but some monocots still keep this in their genome.

Response: We thank the reviewer raising this point and agree that it requires modifications. As the reviewer suggested, we do phylogenetic analysis of MPL1 homologs again by adding some Banana species. The results showed that MPL1, PALM1, and POP form a distinct clade closely related to the SUP/SRO and RBE/St clades (Supplementary Fig. 8a (below figure 1)). The PALM1/POP/MPL1 proteins are highly conserved in all eudicots and magnoliids, while show relatively greater divergence in other land plants, and are even lost in some lineages, such as rice and maize (Fig. 3f (below figure 2); Supplementary Fig. 8b). We have revised the corresponding section.

Figure 1

f

Figure 2